# "I did not know it was a medical condition": Predictors, severity and help seeking behaviors of women with female sexual dysfunction in the Volta region of Ghana

Bolade Ibine[1,2], Linda Sefakor Ametepe[1], Maxfield Okere[3], Martina Anto-Ocrah[4,5,6]*

1 Obstetrics and Gynecology, University of Health and Allied Sciences, School of Medicine, Ho, Volta Region, Ghana, 2 Department of Obstetrics and Gynecology, Family Health Medical School, Accra, Ghana, 3 Department of Biostatistics, Korle Bu Teaching Hospital, Accra, Greater Accra Region, Ghana, 4 Department of Emergency Medicine University of Rochester School of Medicine and Dentistry, Rochester, New York, United States of America, 5 Department of Obstetrics and Gynecology, University of Rochester School of Medicine and Dentistry, Rochester, New York, United States of America, 6 Department of Neurology, University of Rochester School of Medicine and Dentistry, Rochester, New York, United States of America

* martina_anto-ocrah@urmc.rochester.edu

**Data Availability Statement:** All relevant data are within the paper and its Supporting Information files.

## Abstract

### Objectives

The study's main objective was to describe the prevalence and severity of female sexual dysfunction (FSD) amongst a group of Ghanaian women in the outpatient setting of the predominantly rural Volta region of Ghana. Additionally we determine the predictors of FSD severity and care seeking behaviors of women with the condition.

### Study design and setting

This was a cross sectional study conducted in the outpatient setting of the Ho Teaching Hospital in the rural-savannah, agro-ecological zone of Volta Region, Ghana.

### Methods and procedures

FSD was assessed using the Female Sexual Function Index (FSFI) questionnaire. FSD was defined with a cutoff of ≤23 so as not to under-estimate the prevalence in this conservative setting. FSFI score >23 was designated "no FSD". We further categorized women with FSD as having mild (FSFI Total score 18–23), moderate (FSFI Total score <18 to >10) or severe (FSFI Total score ≤10) FSD. Due to sample size restrictions, we combined the moderate and severe FSD groups in our analyses and defined "moderate/severe FSD" as an FSFI Total score < 18. Participants with FSD were further asked to indicate whether or not they sought help for their conditions, the reasons they sought help, and the types of help they sought. We used $p < 0.05$ to determine statistical significance for all analyses and logistic regression models were used to determine crude and age-adjusted effect estimates.

**Funding:** The author(s) received no specific funding for this work.

**Competing interests:** The authors have declared that no competing interests exist.

## Results

FSD Prevalence: Out of 407 women approached, 300 (83.8%) agreed and consented to participate in the study. The prevalence of FSD was 48.3% (n = 145). Compared to those without FSD, over a third of the FSD women resided in rural settings (37.90% vs 20.60%; p = 0.001) and tended to be multiparous, with a significantly greater proportion having at least three children (31.70% vs 18.10%; p = 0.033).

FSD Severity: Over a quarter of the sample (27.6%, n = 40) met the cut-off for moderate to severe FSD. In age-adjusted models, lubrication disorder was associated with 45 times the odds of moderate/severe FSD (age-adj. OR: 45.38, 95% CI: 8.37, 246.00; p<0.001), pain with 17times the odds (age-adj. OR: 17.18, 95% CI: 4.50, 65.50; p<0.001) and satisfaction almost 5times the odds (age-adj. OR: 4.69, 95% CI: 1.09, 20.2; p = 0.04). Compared to those with 1–3 children, nulliparous women had 3.5 times higher odds of moderate/severe sexual dysfunction as well (age-adj. OR:3.51, 95% CI:1.37,8.98; p = 0.009).

FSD-related Health Seeking Behaviors: Statistically significant predictors of FSD-related care seeking included having FSD of pain disorder (age-adj. OR: 5.91, 95% CI:1.29, 27.15; p = 0.02), having ≥4 children (age-adj. OR: 6.29, 95%CI: 1.53, 25.76; p = 0.01). Of those who sought help, seven in 10 sought formal help from a healthcare provider, with General Practitioners preferred over Gynecologist. About one in 3 (31.3%) who did not seek help indicated that they did not know their sexual dysfunction was a medical condition, over a quarter (28.9%) "thought it was normal" to have FSD, and interestingly, 14.1% did not think a medical provider would be able to provide them with assistance.

## Conclusions

Sexual dysfunctions are prevalent yet taboo subjects in many countries, including Ghana. Awareness raising and efforts to feminize the physician workforce are necessary to meet the healthcare needs of vulnerable members of Ghanaian society.

## Introduction

The World Health Organization defines sexuality as a state of physical, emotional, mental and social well-being; a central aspect of "being human" that encompasses the possibility of having pleasurable and safe sexual experiences [1, 2]. Although sexual functioning is an essential aspect of human life, sexual problems are pervasive, and can result in severe consequences for the individual (and potentially their partner), if not addressed. Sexual dysfunction, defined as difficulty experienced by an individual or a couple during any stage of a normal sexual activity, including physical pleasure, desire, preference, arousal or orgasm [3, 4] has been associated with depression and other common mental disorders, relationship discord, poor self-rated health, infertility and overall quality of life [4–11]. Per the published literature, the prevalence of sexual dysfunctions is extremely high, and ranges from 10–63% across various global populations [7, 11–14].

Sexual problems are often multifactorial and can be classified into four broad categories, which are: biomedical (biological factors such as pregnancy, injury and disability, cardiovascular and other chronic diseases that can interfere with intercourse); intrapsychic (psychological elements within the individual, that influence their sexual expression e.g., values learned about

one's body, nudity, "where babies come from", puberty, etc); interpersonal (factors that influence one's ability to engage in sexual relationships, such as communication difficulties with a partner, the extent to which partners share compatible visions of sex, eroticism, pleasure etc.); and socio-cultural/economic/political ('blueprints' for sexual norms, beliefs, values, practices and attitudes that are created and imposed on an individual based on the 'moral code' of their society. Includes religion.) [4, 15]. These four factors often overlap, and rarely occur in isolation to cause the sexual dysfunction [4]. Though sexual dysfunctions are common in both sexes, the greatest morbidity has been reported in women, particularly those in the African region [7]. In many African cultures, the discussion of sexual issues is generally considered a taboo [15, 16]; with conversations of sexuality heavily focused on religion, "moral behavior" and abstinence [15]. The mere mention of sex is often synonymous with deviant behavior [15], suppressing discussions of sexuality as a whole, and sexual dysfunctions in particular. Female sexual dysfunction (FSD), defined as persistent or recurring decrease in a woman's sexual desire, sexual arousal, painful sex (dyspareunia) and/or difficulty in or inability to achieve orgasm [7, 17], is rarely acknowledged in many societies and cultures [7, 18, 19]; though emerging literature suggests that the condition is quite common amongst African women in particular [7, 20, 21]. In the West African country of Ghana, the prevalence of FSD has been estimated to be as high as 72% [20, 21], 30% higher than the global prevalence [7]. These estimates however, are based on mostly urban heterosexual populations [21], limiting their generalizability to women in rural settings, who may be most vulnerable to adverse gynecological and mental health outcomes [22–25]. Moreover, the landmark evaluation of FSD amongst Ghanaian women [20], was based on a subpopulation of female urology patients, a clinical subgroup that would have greater predisposition for, and higher awareness of the dysfunction [26] than the average Ghanaian woman. Thus current prevalence estimates of FSD in Ghana may have some inherent biases.

The government of Ghana and the Ministry of Health, as part of the national agenda, have adopted sexual health as a component of reproductive health [16, 20]. Emphasis, however, has often been placed on sexually transmitted infections and contraception use, with little attention to, or a total neglect for sexual dysfunctions (SD) [20, 27]. In this paper, we describe the prevalence and severity of FSD amongst a group of Ghanaian women in the outpatient setting of the predominantly rural Volta region of Ghana. We describe the predictors of FSD severity and the care seeking behaviors of women with the condition; in the hopes of identifying possible opportunities for developing contextually appropriate interventions for this subpopulation of women.

## Methods

### Study Setting

Ghana is a West African country with a population of 26.7 million and an annual gross domestic product (GDP) per capita income of $4,100 [28–31]. With a land mass of 92,099 square miles, the country is comparable in size to the United Kingdom, with its' western border delineated by the Ivory Coast, the north by Burkina Faso, east by Togo, and the Atlantic Ocean covering its' southernmost borders [31]. English is the official language, even though Ghanaians and non-Ghanaians of various ethnic, linguistic and religious backgrounds call the country home [28, 32, 33]. Similar to the 50 American states, Ghana is currently divided into 16 regions, with 45% of the population spread over large and rural expanses of land [28, 30, 34–36]. Volta Region, the site of the study, is considered part of the 'rural-savannah' agro-ecological zone of the country. The region is the farthest east, and shares boundaries with the Republic of Togo. Over a third of the residents (37.9%) are in rural settings and agriculture and fishery

account for 71% of employment [36]. Volta is one of Ghana's 16 administrative regions, with Ho designated as its capital. The total population of the Ho capital is estimated at 177,281, which represents 8.4% of the region's total population. Females constitute 53% of the population, and an estimated 26.5% of the 11 and over population have no formal education compared to the national rate of 25.9%; with literacy rates for males being higher than that of females [37, 38].

A convenient sample of women in the outpatient setting of the Ho Teaching Hospital were recruited for this study. The Ho Teaching Hospital is a government-owned, 340-bed capacity hospital, equipped to provide services for several out-and in-patient departments including (but not limited to) surgery, medicine, obstetrics and gynaecology, diagnostics, paediatrics and psychiatry.

## Eligibility criteria

All women were encouraged to participate in this cross-sectional study, irrespective of their relationship status. To be eligible, participants had to be consenting adults over the age of 18years, who were seeking care in the outpatient setting of the Ho Teaching Hospital during the period of data collection, which spanned April 17th 2019 to May 22nd 2019. Women who were admitted, or had major mental or cognitive impairment, speech impairment and major morbidity like stroke and psychiatric illness were excluded from the study, as were those who refused consent. The study was approved by the University of Health and Allied Sciences' Research Ethics Committee, with permission obtained from the Ho Teaching Hospital administration. As specified and approved by the University of Health and Allied Sciences' Research Ethics Committee, we obtained written consent from all participants to conduct this study. All consent forms were kept separately from the de-identified survey questionnaires, in a secure location under the supervision of the final author. Keeping the signed consents separate from the survey questionnaires was to ensure that participants' responses could not be connected to their identities.

## Sample size determination

Based on a prevalence estimate of 72%, an alpha of 0.05 and a 95% confidence interval, an estimated sample size of 301 was required to complete the study [39].

## Data collection instrument & measures

**Demographics.** A standard questionnaire (see Appendix) was used to collect socio-demographic data about participants' demographic (age, educational attainment, employment status, religious affiliation, residential setting), relational (marital status, parity), and recruitment (i.e. outpatient department recruited from e.g. surgery, medicine, obstetrics and gynaecology, etc.) characteristics.

**Sexual functioning.** Female sexual function was assessed using the 19-item Female Sexual Function Index (FSFI) questionnaire [11, 40, 41]. With an internal consistency of 0.89–0.96, the FSFI is the most frequently used questionnaire for assessing FSD in various populations across the literature [11, 41]. The measure has been deemed reliable for Ghanaian populations, and used previously in clinical, community and population-based assessments of sexual functioning in Ghanaian women [18, 21]. The FSFI evaluates six domains of female sexual functioning: desire (2 items:Q1,Q2), arousal (4 items:Q3-Q6), lubrication (4 items: Q7-Q10), orgasm (3 items: Q11-Q13), satisfaction (3 items: Q14-Q16) and pain (3 items: Q17-Q19) during sexual intercourse. Study participants are asked to choose the response option that best suited their sexual function during the past four weeks. Scores for items 3–14 and 17–19 range

from 0 ("No Sexual Activity") to 5 ("Very high" or "Almost always/ Always"), and for items 1, 2, 15 and 16, range from 1 ("Almost never/Never" or "Very low/None at all") to 5 ("Very high/ satisfied" or "Almost always/Always"). Each domain is ascribed a factor (0.6 for desire, 0.3 for arousal & lubrication, and 0.4 for orgasm, satisfaction and pain) [11, 40, 41]. The scores for each domain are calculated by adding the scores of the individual domain items and multiplying the sum by the domain factor. Total score is obtained by adding the six domain scores. The full-scale score range is from 2.0 to 36.0, with lower scores indicative of poor sexual functioning. Since the typical cutoff of FSFI score of $\leq 26.55$ for FSD is less sensitive in conservative settings, we used an FSFI score $\leq 23$ to designate FSD [11], so as not to underestimate the FSD prevalence. Thus women with FSD had an FSFI Total score of 23 or lower, and "no FSD" was designated as an FSFI Total score $>23$. We further used the following criteria to categorize FSD severity [11]: mild FSD (FSFI Total score 18–23), moderate FSD (FSFI Total score $<18$ to $>10$), severe FSD (FSFI Total score $\leq 10$). Due to sample size restrictions, we combined the moderate and severe FSD groups in our analyses and defined "moderate/severe FSD" as an FSFI Total score below 18.

**Help seeking behaviors.** Participants with FSD were further asked to indicate whether or not they sought help for their sexual dysfunction, the reasons they sought help, and the types of help sought. Help sought from medical providers were designated as "formal help" whereas "Informal help" was ascribed to friends, family and prayer camps. Responses of "Other" were prompted for further elaboration.

All surveys were interviewer led and completed in-person (i.e. face-to-face) with each respondent. A translated version of the questionnaire was used for those who were not English speaking. The questionnaire was translated into the local dialect, pre-tested to ensure accuracy of translation, and administered by trained study personnel. All interviews were conducted in a screen-protected and discrete study-designated area of the outpatient departments of the Ho Teaching Hospital where the study was conducted. Though there are many languages spoken in the Volta region, from our training and pre-testing phases preparatory to the study, we found most respondents could comfortably speak Ewe, Twi, English or combinations of these. We pretested the survey questionnaire in the following two steps: i) Interviewing of random respondents within the Ho Teaching Hospital to determine participants' language preferences and to ensure uniformity among study personnel, and 2) under the supervision of the senior author (BI), interviewers practiced administering the survey to ensure consistency of delivery and translation of questions by all study personnel.

**Statistical analyses.** We considered in our analyses, several covariates that are associated with sexual functioning, including age, relationship status, parity, education, religious affiliation and residential setting (urban/rural). Univariate analyses and descriptive statistics were used to examine the frequencies and distributions of the study population by FSD group (Has FSD/No FSD), domain (desire, arousal, lubrication, orgasm, satisfaction, pain) and severity (mild, moderate, severe). Categorical variables were compared using chi-square $(X^2)$ or Fisher's Exact tests, where appropriate, while comparisons of continuous outcomes were done using t-tests and Anova estimates, with a parametric distribution of FSD in the study sample.

Age was found to be associated with several other covariates and predictors, and was the most significant predictor of FSD outcome in this study. Adjusting for age only in logistic regression analyses gave us the most parsimonious models. Thus in our logistic regression models, we present both crude and age-adjusted effect estimates. All computations were done at the 95% confidence level, and we report odds ratios and their associated 95% confidence intervals. We used $p < 0.05$ to determine statistical significance for all analyses.

## Results

### Demographics and FSD prevalence

Of 407 women approached, 341 (83.8%) agreed and consented to participate in the study. Fourty one (12%) had incomplete questionnaires, where subjects declined further participation because they found the questionnaire exhausting or intrusive or because of time constraints. The attributes of the final 300 subjects are presented in Table 1. The majority of participants were recruited from the Gynecological and General Outpatient departments of the Ho Teaching Hospital. As depicted, the prevalence of FSD was 48.3% (n = 145), with women who endorsed the dysfunction being significantly older than those without the dysfunction (mean age 34.0(+/-9.5) versus 29.3(+/-6.7) years; p<0.001). Of the women with FSD, the majority (73.8%) were below the age of 40, with most (42.1%) spanning ages 30 and 39 years, and about a third (31.7%) between ages 18 and 29. Approximating the rural population of the Volta Region, over a third of the FSD women resided in rural settings (37.9% vs 20.6% p = 0.001) and tended to be multiparous, with a significantly greater proportion having at least three children (31.7% vs 18.1; p = 0.033), compared to those without FSD.

There were also significant differences in education between the groups. Three out of every 25 women with FSD (12.4%) had no formal education, compared to 1 in 25 (3.9%) uneducated women in the "No FSD" group; and the proportion of those with tertiary education was also lower for FSD women compared to those without the dysfunction (31.7% vs 50.3%; p = 0.002). The employment rates for women in the FSD group was slightly higher than those without the dysfunction, though the difference was not statistically significant (77.9% vs 70.3%; p = 0.148).

Amongst women with FSD, the most commonly endorsed domains were Desire (97.2%), Orgasm (82.8%) and Arousal (80.0%), with one in five (20%) women endorsing dysfunctions in all 6 domains. Forty women (27.6%), representing over a quarter of the sample, met the cut-off for moderate to severe FSD. Women with severe FSD were on average, significantly younger than the rest of the group (Fig 1). Six in 10 (63.6%) women with severe FSD was between the ages of 18 and 29; and 91% of all women with severe FSD were under the age of 40 (Fig 1). These age distributions were significantly greater than any of the others groups'.

Age was found to be associated with several other covariates and predictors, and was the most significant predictor of FSD outcome in this study. Adjusting for age only in logistic regression analyses gave us the most parsimonious models, thus Table 2 shows crude and age-adjusted effect estimates for the predictors of FSD severity.

### FSD severity

In both crude and age-adjusted models, women with lubrication, pain and satisfaction disorders had statistically greater odds of reporting moderate/severe FSD (Table 2). Lubrication disorder was associated with 45 times the odds of moderate/severe FSD (95% CI: 8.37, 246.00; p<0.001) whilst pain (OR: 17.18, 95% CI: 4.50, 65.50; p<0.001) and satisfaction (OR: 4.69, 95% CI: 1.09, 20.20; p = 0.04) were lower in comparison, but highly significant. Compared to those with 1–3 children, women at both ends of the parity spectrum had higher odds of moderate/severe sexual dysfunction; though the effect estimate for those with no children (OR: 3.51; 95% CI: 1.37, 8.98; p = 0.009) was much greater than those with 4 or more offspring (OR:1.64; 95% CI: 0.61, 4.39; p = 0.32). It is important to note that 21.0% (n = 19) of women with no children were married.

Lower educational attainment and residing in a rural setting were, surprisingly, protective for moderate/severe FSD. At all levels of education, FSD women without tertiary training had consistently lower odds of reporting moderate/severe sexual dysfunction compared to those

**Table 1. Demographic and FSD attributes of study population (n = 300).**

| | Has FSD FSFI score ≤23) (n = 145, 48.3%) | No FSD FSFI score >23 (n = 155, 51.7%) | p value | Total n = 300 |
|---|---|---|---|---|
| **Demographic Attributes (n,%)** | | | | |
| **Age, in years** mean (+/-SD) | 34.0 (+/-9.5) | 29.3 (+/-6.7) | **<0.001** | 31.6 (+/-8.5) |
| 18–29 | 46 (31.7) | 80 (51.6) | | 126 (42.0) |
| 30–39 | 61 (42.1) | 61 (39.4) | **<0.001** | 122 (40.7) |
| ≥40 | 38 (26.2) | 14 (9.0) | | 52 (17.3) |
| Range | 18–62 | 18–49 | | 18–62 |
| **Relationship Status** | | | 0.206 | |
| Married | 90 (62.1) | 92 (59.4) | | 182 (60.7) |
| Single | 38 (26.2) | 52 (33.5) | | 90 (30.0) |
| Co-habitating | 17 (11.7) | 11 (7.1) | | 28 (9.3) |
| **Parity** mean (+/-SD) | 1.9 (+/-1.5) | 1.3 (+/-1.2) | **<0.001** | 1.6 (+/-1.4) |
| 0 | 37 (25.5) | 53 (34.2) | | 90 (30.0) |
| 1 | 23 (15.9) | 33 (21.3) | **0.033** | 56 (18.7) |
| 2 | 39 (26.9) | 41 (26.5) | | 80 (26.7) |
| ≥3 | 46 (31.7) | 28 (18.1) | | 74 (24.7) |
| Range | 0–7 | 0–5 | | 0–7 |
| **Educational Attainment** | | | **0.002** | |
| No formal Education | 18 (12.4) | 6 (3.9) | | 24 (8.0) |
| Basic Education (Primary Grades 1–6) | 42 (29.0) | 36 (23.2) | | 78 (26.0) |
| Secondary Education (Junior/Senior High School grades 7–12) | 39 (26.9) | 35 (22.6) | | 74 (24.7) |
| Tertiary Education (University and Higher) | 46 (31.7) | 78 (50.3) | | 124 (41.3) |
| **Religious Affiliation** | | | 0.406 | |
| Christian | 139 (95.9) | 147 (94.8) | | 286 (95.3) |
| Muslim | 5 (3.4) | 4 (2.6) | | 9 (3.0) |
| Other | 1 (0.7) | 4 (2.6) | | 5 (1.7) |
| **Employment status** | | | 0.148 | |
| Unemployed | 32 (22.1) | 46 (29.7) | | 78 (26.0) |
| Employed | 113 (77.9) | 109 (70.3) | | 222 (74.0) |
| Self-employed | 42 (37.2) | 60 (55.0) | | 102 (34.0) |
| Employed by others | 71 (62.8) | 49 (45.0) | | 120 (40.0) |
| **Residential Setting** | | | **0.001** | |
| Urban setting | 90 (62.1) | 123 (79.4) | | 213 (71.0) |
| Rural setting | 55 (37.9) | 32 (20.6) | | 87 (29.0) |
| **Outpatient Setting recruited from** | | | **0.001** | |
| Gynecological | 63 (43.4) | 56 (36.1) | | 119 (39.7) |
| General | 40 (27.6) | 20 (12.9) | | 60 (20.0) |
| Surgical | 16 (11.0) | 23 (14.8) | | 39 (13.0) |
| Dental | 12 (8.3) | 16 (10.3) | | 28 (9.3) |
| Ear/Nose/Throat | 7 (4.8) | 17 (11.0) | | 24 (8.0) |
| Eye Clinic | 7 (4.8) | 23 (14.8) | | 30 (10.0) |
| **FSD Attributes (n,%)** | | | | |
| **FSD Domain Prevalence** | | | | |

*(Continued)*

**Table 1.** (Continued)

| | Has FSD FSFI score ≤23 (n = 145, 48.3%) | No FSD FSFI score >23 (n = 155, 51.7%) | p value | Total n = 300 |
|---|---|---|---|---|
| Desire | 141 (97.2) | - | | 141 (47.0) |
| Arousal | 116 (80.0) | - | | 116 (37.8) |
| Lubrication | 74 (51.0) | - | | 74 (24.7) |
| Orgasm | 120 (82.8) | - | | 120 (40.0) |
| Satisfaction | 112 (77.2) | - | | 112 (37.3) |
| Pain | 84 (57.9) | - | | 84 (28.0) |
| | **Has FSD FSFI score ≤23 (n = 145, 48.3%)** | **No FSD FSFI score >23 (n = 155, 51.7%)** | **p value** | **Total n = 300** |
| **Number of FSD Domains endorsed** | | | | |
| One Domain | 0 (0.0) | - | | 0 (0.0) |
| 2 Domains | 5 (3.4) | - | | 5 (1.7) |
| 3 Domains | 23 (15.9) | - | | 23 (7.7) |
| 4 Domains | 46 (31.7) | - | | 46 (15.3) |
| 5 Domains | 42 (29.0) | - | | 42 (14.0) |
| All 6 Domains | 29 (20.0) | - | | 29 (9.7) |
| **FSD Severity** | | | | |
| Mild FSD (FSFI score 18 to 23) | 105 (72.4) | - | | 105 (35.0) |
| Moderate FSD (FSFI score <18 to >10) | 29 (20.0) | - | | 29 (9.7) |
| Severe FSD (FSFI score ≤10) | 11 (7.6) | - | | 11 (3.7) |
| Moderate/Severe FSD (FSFI score <18) | 40 (27.6) | | | 40 (13.3) |
| **Help Seeking Behaviors** | | | | |
| Sought Help for Sexual Dysfunction | 17 (11.7) | - | | 17 (5.7) |
| Did not Seek Help | 128 (88.3) | - | | 128 (42.7%) |

with university level education or higher. Residing in a rural setting was also associated with lower odds of moderate/severe FSD, though residential setting was not a significant predictor of the outcome.

## Care seeking behaviors

Out of the 145 women with FSD, 17 (11.7%) sought help for their dysfunction (Table 1). Statistically significant predictors of FSD-related care seeking, depicted in Table 3, included dissatisfied (age-adj. OR: 0.30, 95% CI: 0.10, 0.86; p = 0.03) or painful sex (age-adj OR: 5.91, 95% CI:1.29, 27.15; p = 0.02), having 4 more children (age-adj OR: 6.29; 95% CI: 1.53, 25.76; p = 0.01), or residing in a rural setting (age-adj. OR: 2.87, 95%CI: 1.00, 8.24; p = 0.05).

Eight in ten women (82.4%) were prompted to seek help due to the emotional strain of the sexual dysfunction, 12% cited relationship strain as a prompt for their care-seeking, and one woman (5.8%) sought help upon the advice of her pastor (Table 3). Of those who sought help, seven in 10 sought formal assistance from a healthcare provider, with the majority (83%) consulting a General Practitioner for their dysfunctions; and only 16.7% seeking the care of a Gynecologist.

Of the 128 women with FSD who did not seek help, about one in 3 (31.3%) indicated that they did not know their sexual dysfunction was a medical condition. Over a quarter (28.9%) "thought it was normal" to have FSD, and interestingly, 14.1% did not think a medical provider would be able to provide them with assistance.

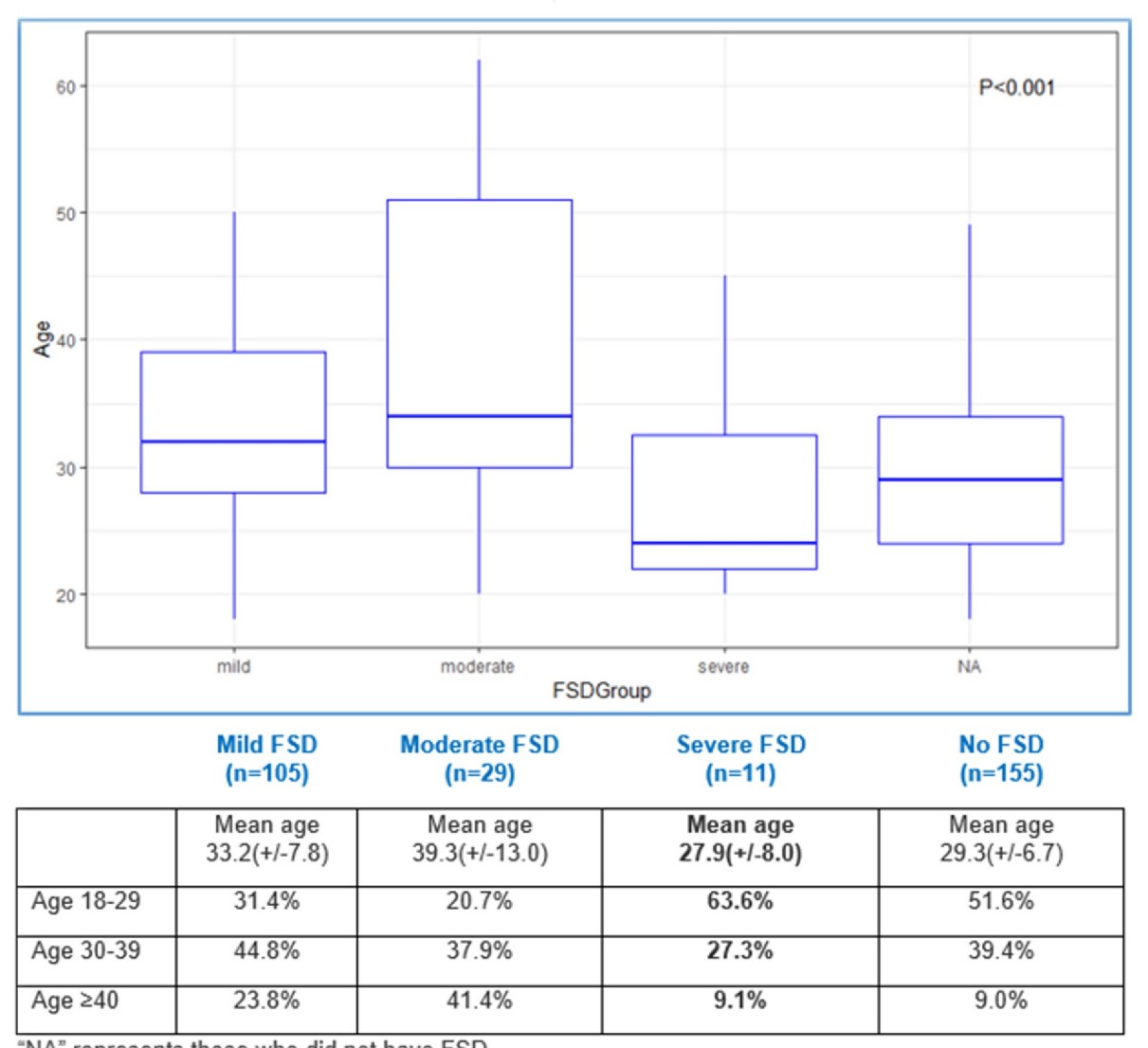

| | Mild FSD (n=105) | Moderate FSD (n=29) | Severe FSD (n=11) | No FSD (n=155) |
|---|---|---|---|---|
| | Mean age 33.2(+/-7.8) | Mean age 39.3(+/-13.0) | Mean age 27.9(+/-8.0) | Mean age 29.3(+/-6.7) |
| Age 18-29 | 31.4% | 20.7% | 63.6% | 51.6% |
| Age 30-39 | 44.8% | 37.9% | 27.3% | 39.4% |
| Age ≥40 | 23.8% | 41.4% | 9.1% | 9.0% |

"NA" represents those who did not have FSD.

**Fig 1. Boxplot of mean age by FSD severity, p<0.001.**

## Discussion

Though human sexual function has requisite biological underpinnings, it is usually experienced as a complex interaction of biological, intrapsychic, interpersonal and socio-cultural/economic/political factors [4, 15]. In many parts of the world, sexuality is considered a sensitive topic, and sexual discussions-whether they be of displeasure or not, are discouraged. Female sexuality is especially taboo; particularly in religious countries like Ghana. Thus women with pathological disorders in sexual functioning may not seek and/or receive the diagnosis and treatment that they may need for their clinically diagnosable and treatable sexual disorders. This is the first study that to our knowledge, evaluates the prevalence, severity and help-seeking behaviors of women with FSD in the mostly rural and agrarian setting of the Volta region of Ghana. Almost half of the women in our study met the cutoff for female sexual dysfunction. The FSD prevalence estimate of 48.3% in our study, approximates the global

**Table 2. Predictors of Moderate/severe sexual dysfunction amongst women with female sexual dysfunction (n = 145).**

| Moderate/Severe FSD[†] (FSFI Score <18) | Crude Odds Ratio | Age-Adjusted Odds Ratio |
|---|---|---|
| **FSD Domain** | | |
| Desire Disorder (vs Not) | 2.34 (95% CI: 0.07, 75.30; p = 0.63) | 1.89 (95% CI: 0.06, 57.80; p = 0.72) |
| Arousal Disorder (vs Not) | 9.45 (95% CI: 0.74, 120.60; p = 0.08) | 8.87 (95% CI: 0.70, 113.0; p = 0.09) |
| Lubrication Disorder (vs Not) | 49.52 (95%CI: 9.27, 264.50; p<0.001) | **45.38 (95%CI: 8.37, 246.00; p<0.001)** |
| Orgasm Disorder (vs Not) | n/a | n/a |
| Satisfaction Disorder (vs Not) | 4.85 (95% CI: 1.15, 20.50; p = 0.03) | **4.69 (95% CI: 1.09, 20.20; p = 0.04)** |
| Pain Disorder (vs Not) | 15.81 (95%CI: 4.30, 58.20; p<0.001) | **17.18 (95%CI: 4.50, 65.50; p<0.001)** |
| **Parity** | | |
| 0 children | 2.42 (95% CI:1.04, 5.65; p = 0.04) | **3.51 (95% CI: 1.37, 8.98; p = 0.009)** |
| 1–3 children | ref | ref |
| ≥4 children | 1.84 (95% CI: 0.70, 4.82; p = 0.22) | 1.64 (95% CI: 0.61, 4.39; p = 0.32) |
| **Education** | | |
| No formal Education | 0.46 (95% CI: 0.14, 1.50; p = 0.20) | **0.27 (95% CI: 0.07, 0.97; p = 0.045)** |
| Basic Education (Primary Grades 1–6) | 0.13 (95% CI: 0.04, 0.41; p = 0.001) | **0.09 (95% CI: 0.02, 0.31; p<0.001)** |
| Secondary Education (Junior/Senior High Schoolgrades 7–12) | 0.41 (95% CI: 0.16, 1.03; p = 0.06) | **0.31 (95% CI: 0.11, 0.83; p = 0.02)** |
| Tertiary Education (University and Higher) | ref | ref |
| **Residential Setting** | | |
| Urban | ref | ref |
| Rural | 0.84 (95% CI: 0.39, 1.80; p = 0.66) | 0.74 (95% CI: 0.34, 1.62; p = 0.45) |

[†]n = 40, using mild FSD (n = 105) as reference

n/a = unable to compute due to zero cells

prevalence of 40.9% [7] and is similar to rates reported for non-clinical, mostly urban Ghanaian females in heterosexual relationships (45.6%) who hailed from the Greater Accra, Ashanti, Western, Central, Brong-Ahafo and Eastern regions. [21]. Though often shrouded in taboo and stigma, research shows that sexual and reproductive health is fundamental to the economic development of nations, to people's health and survival, and to the wellbeing of humanity [42]. Compared to those in urban settings and of higher socio-economic status, women in rural settings, and those of lower socio-economic standing, tend to be more vulnerable to adverse physical, mental, obstetric and gynecological health outcomes [22, 43–53]. They often report poorer self-rated mental and physical health [43, 52], have less access to necessary obstetric care [22, 46, 47, 50], and have higher rates of trauma and mortality [23, 24, 46, 54–59]. Poor sexual functioning may further exacerbate these and other adverse health conditions in this subpopulation. Not surprisingly, women with FSD in our study were most likely to be rural residents, have lower educational attainment and on average, have more children than those without sexual dysfunctions. Given the associations between sexual health, economic growth and overall population health, future studies should engage government stakeholders and policy makers to incorporate sexual dysfunctions as a core component of the government's reproductive health agenda. The Ministry of Health's reproductive health policy could

**Table 3. Predictors and help seeking characteristics of women with female sexual dysfunction (n = 145)[†].**

| | Crude Estimates | Age-Adjusted Estimates |
|---|---|---|
| **FSD Severity** | | |
| Moderate/Severe FSD (vs Mild) | 2.02 (95%CI: 0.71,5.72; p = 0.19) | 2.17 (95%CI: 0.75,6.29; p = 0.16) |
| **FSD Domain** | | |
| Desire (vs Not) | n/a | n/a |
| Arousal Disorder (vs Not) | 0.77 (95% CI: 0.17,3.51; p = 0.74) | 0.79 (95% CI: 0.23, 2.69; p = 0.70) |
| Lubrication Disorder (vs Not) | 2.33 (95%CI:0.69, 7.79; p = 0.17) | 2.90 (95% CI: 0.97, 9.20; p = 0.06) |
| Orgasm Disorder (vs Not) | 1.18 (95%CI:0.23, 6.00; p = 0.84) | 0.79 (95% CI: 0.20, 3.10; p = 0.74) |
| Satisfaction Disorder (vs Not) | 0.39 (95% CI: 0.13, 1.21; p = 0.10) | **0.30 (95% CI: 0.10, 0.86; p = 0.03)** |
| Pain Disorder (vs Not) | **5.11 (95%CI: 1.09,24.03; p = 0.04)** | **5.91 (95%CI:1.29, 27.15; p = 0.02)** |
| **Number of FSDs Endorsed** | | |
| Endorses >4 FSDs (vs 1–4 FSDs) | 1.20 (95% CI: 0.44,3.30; p = 0.73) | 1.26 (95%CI: 0.45, 3.53; p = 0.66) |
| **Age** | | |
| 18–29 (vs 30–39) | 1.62 (95% CI: 0.54,4.86; p = 0.39) | - |
| ≥40 (vs 30–39) | 0.43 (95% CI: 0.08,2.18; p = 0.31) | - |
| **Parity** | | |
| 0 children (vs 1–3) | **4.05 (95%CI: 1.11,14.80; p = 0.035)** | 2.83 (95%CI: 0.65, 12.29; p = 0.17) |
| ≥4 children (vs 1–3) | **5.05 (95%CI: 1.31,19.50; p = 0.02)** | **6.29 (95%CI: 1.53, 25.76; p = 0.01)** |
| **Education** | | |
| No formal Education (vs Tertiary) | n/a | n/a |
| Basic Education (Primary Grades 1–6) (vs Tertiary) | 0.75 (95% CI: 0.22,2.58; p = 0.65) | 0.86 (95% CI: 0.25, 3.03; p = 0.82) |
| Secondary Education (Junior/Senior High School grades 7–12) (vs Tertiary) | 0.82 (95% CI: 0.24,2.82; p = 0.75) | 0.96 (95% CI: 0.27, 3.39; p = 0.95) |
| **Marital Status** | | |
| Single (vs Married) | 1.50 (95% CI: 0.50,4.47; p = 0.47) | 0.81(95%CI: 0.19,3.42; p = 0.78) |
| Co-habitating (vs Married) | 0.50 (95% CI: 0.06,4.19; p = 0.52) | 0.38 (95% CI: 0.04, 3.39; p = 0.38) |
| **Residential Setting** | | |
| Rural (vs Urban) | 2.64 (95% CI: 0.94,7.39; p = 0.07) | **2.87 (95%CI: 1.00, 8.24; p = 0.05)** |
| **Help Seeking Characteristics (n,%)** | | |
| *Where help was sought (n = 17) | | |
| **Formal** | **12 (70.6%)** | |
| General Practitioner | 10 (83.3%) | |
| Gynecologist | 2 (16.7%) | |
| **Informal** | **4 (23.5%)** | |
| Friends | 3 (75%) | |
| Herbalist | 1 (25%) | |

(*Continued*)

**Table 3.** (Continued)

|  | Crude Estimates | Age-Adjusted Estimates |
|---|---|---|
| No response/Missing | 1(5.9%) |  |
| *Reasons for seeking help (n = 17) |  |  |
| Condition is causing emotional strains | 14 (82.4%) |  |
| Condition is causing relationship/ marriage strains | 2 (11.8%) |  |
| Advised to by relatives/friends/pastor | 1 (5.8%) |  |
| **Reasons for not seeking help (n = 128) |  |  |
| Don't know it is a medical condition | 40 (31.3%) |  |
| Thought it was normal | 37 (28.9%) |  |
| Time constraints | 24 (18.8%) |  |
| Health provider cannot help me | 18 (14.1%) |  |
| Other reason | 1 (0.8%) |  |
| No response/Missing | 8 (6.3%) |  |

[†]n = 17 for sought care, using "did not seek care" (n = 128) as reference

n/a = unable to compute due to zero cells

*asked of those who answered "Yes" to the question "Did you seek for help when you experienced the female sexual dysfunction?" (see Appendix)

**asked of those who answered "No" to the question "Did you seek for help when you experienced the female sexual dysfunction?" (see Appendix)

be expanded beyond the prevention of unwanted pregnancies and sexually transmitted infections [15, 16, 20] to more holistically encourage providers to discuss sexual health and sexual dysfunctions (both male and female forms) with their patients. We acknowledge that such a recommendation would require an enormous psycho-socio-cultural shift across all facets of the country's healthcare system, but

without such a strategy, it will be difficult to recognize the discriminant impact that sexual dysfunctions have on some of the most vulnerable members of Ghanaian society.

It is often assumed that sexual dysfunctions in women are associated with increasing age, the onset of menopause, and several associated hormonal changes [5, 6, 13, 60–65]. However, our results show that FSD is quite common in premenopausal women, particularly those between the ages of 18 and 39 years. Women in this age group were most likely to report that their FSD was severe, with lubrication disorders being the most common disturbance. Compared to those who did not endorse problems with lubrication, women with lubrication disorders had close to 50 times the odds of reporting moderate/severe sexual dysfunction. These findings are similar to those reported by Imbeah et al [21], who reported that 72% of Ghanaian women surveyed in their six region study, cited lubrication difficulties as causes of their dysfunctions. The root cause of FSD for these "young" women may have less to do with age and/ or biological changes in hormonal regulation, and are most likely attributable to the physical/ mechanical stress imposed on the vaginal tissue during sexual encounters. For women, vaginal lubrication is an important part of sex [66]. Lubrication readies the vagina for penetration, and reduces any accompanying friction or irritation during intercourse [4, 66, 67]. Without adequate lubrication, the frictional force imposed on the delicate vaginal tissue can result in bruising and tearing [66], which may lead to infections, painful sexual experiences and overall dissatisfaction with sex. Not surprisingly, lubrication, pain and satisfaction were the three FSD domains most associated with FSD severity in our study. And although sexual pain was the only statistically significant predictor of FSD-associated care seeking behavior, likely at the

root of the woman's dyspareunia is inadequate lubrication. Though lubrication usually occurs naturally, some women become more lubricated than others [66]. It is important that women and their partners understand the role that lubrication plays in comfortable intercourse to reduce the woman's susceptibility for infections, as well as their painful sexual experiences. More studies are needed to evaluate the access, utilization and perception of personal lubricants amongst Ghanaian women and couples.

In our study, Women with FSD often normalized their dysfunctions, with one in three women indicating that they did not know that FSD was a medical condition and 14% noting that a healthcare provider would not be able to resolve their issue. These results again echo the findings of Imbeah and team [21] who indicated that 50%(n = 80) of the women in their study did not seek help because they were under the impression that FSD was normal, and 19.2% (n = 31) were embarrassed to seek help. Through targeted educational efforts, Ghanaian women should be made aware that persistent or recurring decreases in a woman's sexual desire, sexual arousal, painful sex (dyspareunia) and/or difficulty in or inability to achieve orgasm [7, 17] is a medical condition that can be treated by healthcare providers. This awareness raising is necessary, given that 82.4% of those who sought help in our study point to *emotional strain* as their reasons for seeking care for their sexual dysfunction. These women fit the Diagnostic and Statistical Manual of Mental Disorders, Fifth Edition (DSM-5) criteria for the clinical diagnosis of FSD [3, 68–70], and would require referral and treatment by an appropriate healthcare provider. Appropriately, most women in our study sought formal help from a General Practitioner (GP) or a Gynecologist for their conditions.

Ghana, like most countries, is undergoing a gender redistribution of the physician workforce [71–74]. The estimated 1:8,481 patient to provider ratio [75] has traditionally tipped in favor of male doctors [76–79], including Obstetric/Gynecologists (ObGyns) [80], even though there is ample literature to support the notion that women, particularly those in conservative cultures, prefer female OB-GYNs, given the intimacy of the services rendered by these providers [81–83]. Research shows that when patients are treated by physicians "who look like them" or have life experiences that closely match that of their patients', patients perceive their care to be of higher quality, are more satisfied with their care, have greater adherence to medical advice, and overall, have better health outcomes [84–88]. Providers also show a preference for same-gender consultations for sexual health matters [89].The global trend towards feminization of the healthcare workforce promises to bring changes to the patient–physician relationship, the local delivery of care, the global delivery of care to the society as a whole, as well as to the medical profession itself [72, 74]. Female physicians it has been shown, spend more time with their patients [72, 74, 90], write fewer prescriptions and are more attuned to their patients' needs enough to refer more cases to specialists than their male colleagues [72, 74, 91]. The female patient may be more comfortable raising sexual concerns with a female physician. There is however, a glaring male dominance of the ObGyn specialty in Ghana, with less than 10% female representation among practicing gynecologists [92]. Improving the gender balance of the Ghanaian physician workforce would decrease the patient:provider gender mismatch, and potentially improve discussions around the traditionally taboo details of female sexuality. Feminization could result in more women being screened by their GPs, and referred to ObGyns for proper diagnosis and treatment. Gender redistribution of Ghanaian doctors is a necessary step in reducing the "personal embarrassment" associated with women's reluctance to seek help for their sexual dysfunctions, as reported by our study participants. More female doctors are needed to provide care for the majority female Ghanaian population [35] and serve the needs and priorities of the health system. Transitioning to this model of care however, may take a long time to achieve, as it requires not only an attitudinal and cultural shift in the existing healthcare infrastructure, but may also require a dismantling of pre-existing

institutional and structural barriers. Bridging this transition, may be a healthcare model that utilizes Advanced Practice Providers (APPs) such as nurse midwives, who often serve as primary caregivers for patients with obstetric and gynecological needs. Not only is the provider: patient ratio higher for these APPs, but the gender balance between patients and providers is not as disproportional. Nurse midwives could be trained to screen for FSD as part of their "routine" patient care; relieving the workload expectations of other providers.

In spite of these healthcare workforce trends, providers, male or female, should be made aware of and encouraged to discuss sexual health and sexual dysfunctions with their patients. Insufficient assessment of sexual health concerns by Physicians, is at the core of the barriers against appropriate diagnosis and outcome of FSD. Literature has emphasized the inconsistent and avoidant approach by providers to exploring women's sexual concerns [26, 93, 94]. Often times, the burden of reporting embarrassing symptoms and diagnosing of a dysfunction is shifted to the patient [26, 89, 95, 96]. The physician is often unintentionally constrained by multifaceted reasons, prominent among which is confidence in assessment skills. In Ghana, the ObGyn is expected in most cases, to deal with female reproductive and sexual health concerns. Hence the GP and other providers rarely explore detailed sexual history or lack proactivity and social empathy for discussing and identifying clients with sexual health needs. In addition to raising the awareness of patients about the detriments of FSD, there is a corresponding need to improve social empathy, assessment and management skills of Ghanaian physicians, particularly non-ObGyns, who often serve as the first point of contact for patients suffering from the condition.

Because of the social stigma around female sexuality, women tend to avoid and/or are embarrassed to discuss their sexual health with their health care professionals (HCPs) [94]. HCPs-both physicians and APPs-can be trained to improve their empathetic responses to their patients' sexual frustrations by initiating (and maintaining) a sexual health conversation in a manner that is comfortable for the woman to convey her concerns. This initial step is crucial to the trust building necessary for the trained provider to assess the patient, diagnose, treat and manage their FSD [89, 93, 94]. These tools-educating women, training HCPs, and providing communication tools to HCPs-may serve to decrease the stigma associated with sexual health discussions, and facilitate the treatment-oriented dialogue between patients and their providers when dealing with FSDs [94].

## Strengths and limitations

This study, to our knowledge is the first of its kind in the rural-savannah, agro-ecological zone of Ghana's Volta region. Even though most of the study participants were recruited from the Gynecological and General Outpatient departments, we had representation from women in the Surgical, Dental, Ear/Nose/Throat and Ophthalmology departments as well. Our study did not exclude single women, or women who were not in heterosexual relationships, making our findings generalizable to a broad reach of Ghanaian women with varying sexual and relationship experiences. Despite these strengths, our study has a few limitations that bear mentioning. Firstly, restricting the study to women 18 years and older limits the generalizability of our findings to females below this age, particularly teenagers, whose sexual health tends to be understudied in the Ghanaian context. The sexual experiences of the average Ghanaian teenager may be higher and more likely to be deemed "immoral" [19] than that of the "older" female; putting such adolescents at greater risk for gynecological infections, psychological distress and other FSD-related sequelae. More research is needed to better understand the impact of sexual dysfunctions across the spectrum of a Ghanaian woman's reproductive years. Another limitation of our study is in the sample size limitations imposed upon our analyses. Even though the

study was well powered at a prevalence estimate of 72%, the low prevalence of moderate/severe FSD (27.6%, n = 40) resulted in wide and imprecise confidence intervals for the effect estimates in our adjusted analyses. This estimate, as discussed earlier, is likely over-estimated and reflects the prevalence amongst a care-seeking subpopulation of urology patients. The prevalence rate of our sample is in line with other non-clinical Ghanaian populations [18, 21], and also in range with global FSD prevalence estimates [7]. Studies that are powered at the lower prevalence estimates of our findings may yield more robust results from larger studies, that reflect the true population estimates. Such studies are needed to replicate and validate the current findings. Finally, our study does not account for differences in FSD outcomes based on the women's pregnancy status or sexual preferences. Though the FSFI was designed to be used non-discriminately, and has in fact been used to evaluate FSD outcomes in pregnant and non-pregnant heterosexual, lesbian and bisexual populations [13, 97–99], because we do not explicitly compare outcomes within these subgroups in our study, the findings should be interpreted cautiously. As is common in most sexuality studies, our results may be more reflective of a heterosexual majority and/or those with less conservative sexual attitudes and values [4, 100], than that of sexual minorities and/or others with conservative sexual attitudes. This is particularly salient in a country like Ghana, where overt discussions of sexual matters are frowned upon [15]. Those who are most reticent about sexual issues may be most vulnerable to adverse sexual outcomes, and may require greater clinical interventions. Thus future studies should attempt to evaluate FSD outcomes in pregnant Ghanaian women, as well as non-heterosexual individuals, who may have these and other unique challenges. Future research endeavors should also consider qualitative research methodologies that will better contextualize and support the study findings, through the use of narratives and theoretical frameworks. The contribution of this paper may be better appreciated if it were supported with qualitative theories/frameworks/models.

## Conclusions

The study's main objective was to describe the prevalence and severity of female sexual dysfunction (FSD) amongst a group of Ghanaian women in the outpatient setting of the predominantly rural Volta region of Ghana. Additionally we determine the predictors of FSD severity and care seeking behaviors of women with the condition.

We found that FSD is highly prevalent amongst women in rural settings of the Ghanaian community, with lubrication, pain and satisfaction disorders being the most associated with severity. These conditions leave women vulnerable to preventable sexually transmitted infections, physical pain and psychological distress. Most women deem the dysfunctions to be normal, and are unaware that FSD is a clinically diagnosable condition that can be treated by expert physicians. In order to attain the World Health Organization's agenda of "Health for All" by the year 2030 [101, 102], sexual dysfunctions should be incorporated into Ghana's national health agenda to improve the health of some of the most vulnerable members of Ghanaian society.

## Appendix

### DATA COLLECTION TOOL

**FEMALE SEXUAL DYSFUNCTION: PREVALENCE AND HELP- SEEKING BEHAVIOUR AMONG WOMEN ATTENDING THE HO TEACHING HOSPITAL OUT–PATIENT DEPARTMENT**.

**Section A**
**Demographic Data of Participant**

**Part A**

A1. AGE: . . . . . . . . . . . . . . . . . . . . . .

A2. MARITAL STATUS: single (), married (), cohabiting (), deceased ()

A3.RELIGIOUS AFFILIATION: Christianity () Islam () others ()

A4.EDUCATIONAL LEVEL: no education () JHS () High School () Tertiary ()

A5.EMPLOYMENT STATUS: employed () unemployed ()

A6.OCCUPATION. . . . . . . . . . . . . . . . . . . . . . . . . . . . .

A7.AVERAGE MONTHLY INCOME: GHS. . . . . . . . . . ..

A8.NUMBER OF CHILDREN . . . . . . . . . . . . ..

A9.NUMER OF DEPENDANTS. . . . . . . . . . . . . ..

A10.PLACE OF RESIDENCE. . . . . . . . . . . . . . . . . . . . . . ...

A11.CLINIC SPECIALTY (OPD in what

specialty). . . . . . . . . . . . . . . . . . . . . . . . . . . . . . . . . . . .

**Part B**

**B1. Over the past 4 weeks, how often did you feel sexual desire or interest?**

5 = Almost always or always

4 = Most times (more than half the time)

3 = Sometimes (about half the time)

2 = A few times (less than half the time)

1 = Almost never or never

**B2. Over the past 4 weeks, how would you rate your level (degree) of sexual desire or interest?**

5 = Very high

4 = High

3 = Moderate

2 = Low

1 = Very low or none at all

**B3. Over the past 4 weeks, how often did you feel sexually aroused ("turned on") during sexual activity or intercourse?**

0 = No sexual activity

5 = Almost always or always

4 = Most times (more than half the time)

3 = Sometimes (about half the time)

2 = A few times (less than half the time)

1 = Almost never or never

**B4. Over the past 4 weeks, how would you rate your level of sexual arousal ("turn on") during sexual activity or intercourse?**

0 = No sexual activity

5 = Very high

4 = High

3 = Moderate

2 = Low

1 = Very low or none at all

**B5. Over the past 4 weeks, how confident were you about becoming sexually aroused during sexual activity or intercourse?**

0 = No sexual activity

5 = Very high confidence

4 = High confidence

3 = Moderate confidence

2 = Low confidence

1 = Very low or no confidence

**B6. Over the past 4 weeks, how often have you been satisfied with your arousal (excitement) during sexual activity or intercourse?**

0 = No sexual activity

5 = Almost always or always

4 = Most times (more than half the time)

3 = Sometimes (about half the time)

2 = A few times (less than half the time)

1 = Almost never or never

**B7. Over the past 4 weeks, how often did you become lubricated ("wet") during sexual activity or intercourse?**

0 = No sexual activity

5 = Almost always or always

4 = Most times (more than half the time)

3 = Sometimes (about half the time)

2 = A few times (less than half the time)

1 = Almost never or never

**B8. Over the past 4 weeks, how difficult was it to become lubricated ("wet") during sexual activity or intercourse?**

0 = No sexual activity

1 = Extremely difficult or impossible

2 = Very difficult

3 = Difficult

4 = Slightly difficult

5 = Not difficult

**B9. Over the past 4 weeks, how often did you maintain your lubrication ("wetness") until completion of sexual activity or intercourse?**

0 = No sexual activity

5 = Almost always or always

4 = Most times (more than half the time)

3 = Sometimes (about half the time)

2 = A few times (less than half the time)

1 = Almost never or never

**B10. Over the past 4 weeks, how difficult was it to maintain your lubrication ("wetness") until completion of sexual activity or intercourse?**

0 = No sexual activity

1 = Extremely difficult or impossible

2 = Very difficult

3 = Difficult

4 = Slightly difficult

5 = Not difficult

**B11. Over the past 4 weeks, when you had sexual stimulation or intercourse, how often did you reach orgasm (climax)?**

0 = No sexual activity

5 = Almost always or always

4 = Most times (more than half the time)

3 = Sometimes (about half the time)

2 = A few times (less than half the time)

1 = Almost never or never

**B12. Over the past 4 weeks, when you had sexual stimulation or intercourse, how difficult was it for you to reach orgasm (climax)?**

0 = No sexual activity

1 = Extremely difficult or impossible

2 = Very difficult

3 = Difficult

4 = Slightly difficult

5 = Not difficult

**B13. Over the past 4 weeks, how satisfied were you with your ability to reach orgasm (climax) during sexual activity or intercourse?**

0 = No sexual activity

5 = Very satisfied

4 = Moderately satisfied

3 = About equally satisfied and dissatisfied

2 = Moderately dissatisfied

1 = Very dissatisfied

**B14. Over the past 4 weeks, how satisfied have you been with the amount of emotional closeness during sexual activity between you and your partner?**

0 = No sexual activity

5 = Very satisfied

4 = Moderately satisfied

3 = About equally satisfied and dissatisfied

2 = Moderately dissatisfied

1 = Very dissatisfied

**B15. Over the past 4 weeks, how satisfied have you been with your sexual relationship with your partner?**

5 = Very satisfied

4 = Moderately satisfied

3 = About equally satisfied and dissatisfied

2 = Moderately dissatisfied

1 = Very dissatisfied

**B16. Over the past 4 weeks, how satisfied have you been with your overall sexual life?**

5 = Very satisfied

4 = Moderately satisfied

3 = About equally satisfied and dissatisfied

2 = Moderately dissatisfied

1 = Very dissatisfied

**B17. Over the past 4 weeks, how often did you experience discomfort or pain during vaginal penetration?**

0 = Did not attempt intercourse

1 = Almost always or always

2 = Most times (more than half the time)

3 = Sometimes (about half the time)

4 = A few times (less than half the time)

5 = Almost never or never

**B18. Over the past 4 weeks, how often did you experience discomfort or pain following vaginal penetration?**

0 = Did not attempt intercourse

1 = Almost always or always

2 = Most times (more than half the time)

3 = Sometimes (about half the time)

4 = A few times (less than half the time)

5 = Almost never or never

**B19. Over the past 4 weeks, how would you rate your level (degree) of discomfort or pain during or following vaginal penetration?**

0 = Did not attempt intercourse

1 = Very high

2 = High

3 = Moderate

4 = Low

5 = Very low or none at all

**Part c**

**C1.Did you seek for help when you experienced the female sexual dysfunction?**

Yes [] No []

**C2. If no to C1, why didn't you seek for help?**

Health provider cannot help me [] time constraints []

Others. . . . . . . . . . . . . . . . . . . . . . . . . . . . .

**c3. If yes to C1, where did you seek help from?**

Gynecologist () general practitioner () Herbalist () friends () family () Prayer camp ()

Others. . . . . . . . . . . . . . . . . . .

**C4. If yes to C1, why did you seek help?**

Advised to by relatives/friends/pastor [] Referred from a formal healthcare personnel [] condition is causing relationship/marriage strains [] condition is causing emotional strains []

Others. . . . . . . . . . . . . . . . . . . . . . . . . . . . . . . . . . . . . . . . . . . . . . . . . . . . . . . . . . . . . . . . . . . . . . . . .

## Supporting information

**S1 File. Female Sexual Dysfunction Data_Deidentified for submission.**
(ZIP)

## Acknowledgments

Special gratitude goes to Professor Morhe (Head of Department, Department of ObGyn, University of Health and Allied Sciences (UHAS)) for your assistance and guidance throughout the project. From inception, to referencing, to data entry, your input was invaluable. Additional goes to the research assistants and translators who helped with the expeditious and thorough data collection process.

## Author Contributions

**Conceptualization:** Bolade Ibine, Martina Anto-Ocrah.

**Data curation:** Linda Sefakor Ametepe, Maxfield Okere.

**Formal analysis:** Maxfield Okere, Martina Anto-Ocrah.

**Investigation:** Bolade Ibine, Linda Sefakor Ametepe, Martina Anto-Ocrah.

**Methodology:** Bolade Ibine, Martina Anto-Ocrah.

**Project administration:** Bolade Ibine, Linda Sefakor Ametepe.

**Resources:** Bolade Ibine.

**Supervision:** Bolade Ibine, Martina Anto-Ocrah.

**Validation:** Maxfield Okere.

**Visualization:** Martina Anto-Ocrah.

**Writing – original draft:** Martina Anto-Ocrah.

**Writing – review & editing:** Bolade Ibine, Maxfield Okere, Martina Anto-Ocrah.

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
