## [Decision Letter · Decision Letter 0]

9 Sep 2019

PONE-D-19-22745

“I did not know it was a medical condition”: Predictors, Severity and Help Seeking Behaviors of Women with Female Sexual Dysfunction in the Volta Region of Ghana”

PLOS ONE

Dear Dr Anto-Ocrah,

Thank you for submitting your manuscript to PLOS ONE. After careful consideration, we feel that it has merit but does not fully meet PLOS ONE’s publication criteria as it currently stands. Therefore, we invite you to submit a revised version of the manuscript that addresses the points raised during the review process.

We would appreciate receiving your revised manuscript by Oct 24 2019 11:59PM. To enhance the reproducibility of your results, we recommend that if applicable you deposit your laboratory protocols in protocols.io, where a protocol can be assigned its own identifier (DOI) such that it can be cited independently in the future. For instructions see: http://journals.plos.org/plosone/s/submission-guidelines#loc-laboratory-protocols

We look forward to receiving your revised manuscript.

Kind regards,

Joshua Amo-Adjei, Ph.D

Academic Editor

PLOS ONE

Journal Requirements:

1. Please amend your current ethics statement to address the following concerns:  

a) Did participants provide their written or verbal informed consent to participate in this study?

Additional Editor Comments (if provided):

Dear Authors, ensure to pay keen attention to the feedback from Reviewer #1 if you decide to revise and resubmit your manuscript.

Reviewers' comments:

Reviewer's Responses to Questions

**Comments to the Author**

1. Is the manuscript technically sound, and do the data support the conclusions?

Reviewer #1: Partly

Reviewer #2: Yes

2. Has the statistical analysis been performed appropriately and rigorously? 

Reviewer #1: Yes

Reviewer #2: Yes

3. Have the authors made all data underlying the findings in their manuscript fully available?

Reviewer #1: Yes

Reviewer #2: Yes

4. Is the manuscript presented in an intelligible fashion and written in standard English?

Reviewer #1: Yes

Reviewer #2: Yes

5. Review Comments to the Author

Reviewer #1: Dear Authors:

I have read your interesting paper which examines how women view FSD and the care seeking behaviours of women with the condition. I like the topic and I like the paper. I think the paper is underdeveloped. The paper will benefit significantly from a major revision. Some specific comments to improve the paper:

The introductory part of the paper needs to be reworked. The long expose is not connected to the subsequent paragraphs and detracts from texts. The transitioning between the first and second paragraph of the introduction is poor. The authors defined sexuality and straightaway discussed poor sexual functioning without first establishing a link between sexuality and sexual functioning.

One primary concern is how concepts are used very 'vaguely' and not contextualized. Authors failed to tell the reader what is poor/sexual functioning? What do these terminologies mean? What is poor sexual functioning? Can you cite this concept? What does it entail exactly? Do men have poor sexual dysfunction too?

It will be useful to review the factors accounting for sexual dysfunction particularly in the West African context. Is it sociocultural beliefs/practices, anatomical/hereditary, or lifestyle (e.g., alcohol use, smoking, etc.) especially as the central focus of the study was to describe the predictors of FSD severity. Literature is missing on this? Provide some context regarding this.The following publications may help:

Anarfi, J. K., & Owusu, A. Y. 2011. “The Making of a Sexual Being in Ghana: The State, Religion and the Influence of Society as Agents of Sexual Socialization.” Sexuality & Culture 15: 1–18.

Fiaveh, D. Y., Izugbara, C. O., Okyerefo, M. P. K., Reysoo, F., & Fayorsey, C. K. 2015. Constructions of masculinity and femininity and sexual risk negotiation practices among women in urban Ghana. Culture, Health & Sexuality, 17(5), 650-662.

Fiaveh, D. Y., Okyerefo, M. P. K., & Fayorsey, C. K. 2015. Women’s experiences of sexual pleasure in Ghana. Sexuality & Culture, 19(4), 697-714.

The subsection on study setting provides little context to the study. For example, how do population size and per capita income relate to FSD? These were not evident in the demographic attributes presented in Table 1 under the results. The issues that matter (age, relationship/love, educational attainment, religion, employment status, etc.) were not emphasised in the review/introduction. Yet, they appear instrumental in the analysis/results and discussion. For example, the study found a significant relationship between a woman's age and FSD outcome. What does this mean? Would you suggest that younger (say those below age 40) women have different SDs from older women (those above 40 but below 60)?

Some more details about the methodology is required—ethical protocol and procedure/techniques. Was the survey face-to-face? Under what condition where the questionnaires administered? More clarity about the local dialect used since Ewe is not the only local language in the Volta region of Ghana. How were translations/transliterations done? Sexuality is a sensitive issue across ‘borders’ and since the questionnaires were administered by trained study personnel, further details about the processes/techniques could be useful. See:

Fiaveh, D. Y. (2018). Cultural sensitivities: A case study of sexual pleasure in Ghana. Sage Research Methods Cases, 9781526429780. http://dx.doi.org/10.4135/9781526429780.

Another point that deserves much clearer attention and may be an interesting facet of the point above are the measurements— variables. How do relationship status or residential setting or religious affiliation directly impact FSD? And how do we measure these? On the basis of some typology developed, i.e., using a questionnaire? For example, how would different units of analysis in the sample, e.g., pregnant women, non pregnant women, women with kids/children and/or certain number of children, and non-heterosexual women impact the results/claims? How does sexuality operate in Ghanaian society that might silence or closet FSD? Does compulsory heterosexuality (penovaginal 'penetrative' sex) rely upon FSF/D? The FSFI questionnaire has major flaws in 'todays' Ghana. Since the paper is on rural agrarian setting in a surfer of Ghana, a section on limitation will be useful.

The items measured, i.e., desire, arousal, lubrication, orgasm, satisfaction, and pain during sexual intercourse do not suffice to make generalizations. If women desire pleasurable and safe sex, should’t we be discussing/asking what role sexual preference/orientation (e.g., oral sex, use of condom) play in this? What about sex positions, romance, the penis; size, length, girth, etc., for those who prefer 'penetrative' sex play in FSD? If not why not?

The results reflect the methods employed in the data gathered. Narratives were missing and this is ‘sad’. There is need for some qualitative data, interviews, to support positivists’ stance and claims being made in the paper including the proposal to develop contextually appropriate interventions for the subpopulation studied in Ghana. Would the results differ among Akan groups/speakers?

While the discussion is logically organised, it is underdeveloped probably due to the absence of a theory/framework to hold the arguments together. The contribution of this paper will be better appreciated if it were supported with a theory/framework/model.

The part about the link to rural residential setting and implications for FSD (see paragraph 1 line 9-12 under Discussion) is contested and can constitute a sensitive research design including stereotype. Is this a finding or an existing literature? If this is so, this should also be interrogated. This could be due to negative skewness in the sampling technique. You need to engage the data a bit more closely and situate your argument within a paradigm.

There are language problems and sloppiness. The authors need to proofread the paper thoroughly.

All my very best!

Reviewer #2: I am impressed by this manuscript, both by the topic it studies, which I am sure is under studied in all settings, but especially in low resource settings such as Ghana, and by the quality of the paper.

I have just a few comments which are necessary to address before this paper can be published.

1. The methods section doesn't ever state which inferential statistics are performed. The regression analyses need to be explained.

2. Did interviewers only ask the questions about care-seeking to those women who answered in the affirmative about having symptoms of FSD?

3. It is written, "amongst women with FSD, the most commonly endorsed domains were..." What is meant by "endorsed"?

4. Why is Table 3 only unadjusted estimates? Why aren't there multivariate analyses conducted on these data?

5. I would caution the authors against the causal language at the beginning of the discussion section. "This is the first study...that evaluates the predictors...." These are associations, not predictors, since the data are all cross-sectional.

6. The end of the first paragraph in the discussion section clearly lay out the problems that SD poses for women. Is it possible to state a few interventions that could address these? The authors go into some of them later in the discussion section, but it feels like that could be a good place for some information as well.

7. While I do not disagree that a more gender equitable physician workforce would likely help with this, and other, issues (and I appreciate very much the authors going into some detail as to the advantages of having a gender equitable physician workforce), a) that is going to take a very long time to achieve, and b) all physicians *should* be able to address the health concerns of their patients. The authors do say that there is a need to improve "social empathy, assessment and management", but don't say if there are other ways that have proven effective to help physicians help their patients.

8. In addition to what I wrote above, the authors also did not mention nurses or midwives, who are both far more plentiful in the country, and far more likely to be female. Is there a role for nurses and / or midwives in this space?

6. PLOS authors have the option to publish the peer review history of their article (what does this mean?). If published, this will include your full peer review and any attached files.

Reviewer #1: Yes: Daniel Yaw Fiaveh

Reviewer #2: No

---

## [Author Response · Author response to Decision Letter 0]

21 Oct 2019

Editor Comment

Please amend your current ethics statement to address the following concerns: 

a) Did participants provide their written or verbal informed consent to participate in this study?

Response

We have added the following to the Methods section (on page 9):

As specified and approved by the University of Health and Allied Science's Research Review Board, we obtained written consent from all participants to conduct this study. 

Response

Not applicable

Response

Response

We have amended the Ethics Statement with the statement above 

“As specified and approved by the University of Health and Allied Science's Research Review Board, we obtained written consent from all participants to conduct this study. “

 

Reviewer 1

We are truly honored to have had our manuscript reviewed by Dr Yaw Fiaveh, whose work was used to partially inform the current study. Dr Fiaveh, being a qualitative researcher, uses a methodology that is heavily grounded in theory. As providers and implementation scientists, we use approaches that are rooted in quantitative, epidemiological methodologies to attempt to derive practical, clinically-driven solutions to the problem of female sexual dysfunction in the population of interest we surveyed. Our work is often informed by that of qualitative researchers such as Dr Fiaveh, but our interests, motivations, and end goals often diverge; as we, as practitioners, are more driven by patient diagnosing and treatment. 

Bearing these differences in mind, we have attempted to address the concerns raised by Dr Fiaveh as noted below. 

Comment 1

The introductory part of the paper needs to be reworked. The long expose is not connected to the subsequent paragraphs and detracts from texts. The transitioning between the first and second paragraph of the introduction is poor. The authors defined sexuality and straightaway discussed poor sexual functioning without first establishing a link between sexuality and sexual functioning. One primary concern is how concepts are used very 'vaguely' and not contextualized. Authors failed to tell the reader what is poor/sexual functioning? What do these terminologies mean? What is poor sexual functioning? Can you cite this concept? What does it entail exactly? Do men have poor sexual dysfunction too?

Response

We have modified the Introduction as follows (on pages 6-7): 

The World Health Organization defines sexuality as a state of physical, emotional, mental and social well-being; a central aspect of “being human” that encompasses the possibility of having pleasurable and safe sexual experiences [1, 2]. Although sexual functioning is an essential aspect of human life, sexual problems are pervasive, and can result in severe consequences for the individual (and their partner) if not addressed. Sexual dysfunction, defined as difficulty experienced by an individual or a couple during any stage of a normal sexual activity, including physical pleasure, desire, preference, arousal or orgasm [3, 4] has been associated with depression and other common mental disorders, relationship discord, poor self-rated health, infertility and overall quality of life [4-11]. Per the published literature, the prevalence of sexual dysfunctions is extremely high, and ranges from 10-63% across various global populations [7, 11-14]. Sexual problems are often multifactorial and can be classified into four broad categories which are: biomedical (biological factors such as pregnancy, infertility, sexually transmitted infections, injury and disability, cardiovascular and other chronic diseases that can interfere with intercourse); intrapsychic (psychological elements within the individual, that influence their sexual expression e.g., values learned about one’s body, nudity, “where babies come from”, what to expect during puberty, etc); interpersonal (factors that influence one’s ability to engage in sexual relationships, such as communication difficulties with a partner, distrust, power struggles, the extent to which partners share compatible visions of sex, eroticism, pleasure etc.); and socio-cultural/economic/political (‘blueprints’ for sexual norms, beliefs, values, practices and attitudes that are created and imposed on an individual based on the ‘moral code’ of their society. Includes religion, the state and the general established society. Less urban settings may have more conservative beliefs and practices about sexuality) [4, 15]. These four factors often overlap, and rarely occur in isolation to cause the dysfunction [4]. Though sexual dysfunctions are common in both sexes, the greatest morbidity has been reported in women, particularly those in the African region [7]. In many African cultures, the discussion of sexual issues is generally considered a taboo [15, 16]; with conversations of sexuality heavily focused on religion, “moral behavior” and abstinence [15]. The mere mention of sex is often synonymous with deviant behavior [15], suppressing discussions of sexuality as a whole, and sexual dysfunctions in particular. “

Have also added the following to the Discussion (pages 21-22): 

“Human sexuality refers to ‘the sum of the physical, functional, and psychologic attributes that are expressed by one’s gender identity and sexual behavior [15]. Human sexuality goes beyond biological attributes, and includes overlapping constructs intrapsychic, interpersonal and socio-cultural/economic/political factors [4, 15]. In many parts of the world, sexuality is considered a sensitive topic, and sexual discussions-whether they be of displeasure or not, are discouraged. Female sexuality is especially taboo; particularly in religious countries like Ghana. Thus women with pathological disorders in sexual functioning may not seek and/or receive the diagnosis and treatment that they may need for their clinically diagnosable and treatable sexual disorders”.

Comment 2

It will be useful to review the factors accounting for sexual dysfunction particularly in the West African context. Is it sociocultural beliefs/practices, anatomical/hereditary, or lifestyle (e.g., alcohol use, smoking, etc.) especially as the central focus of the study was to describe the predictors of FSD severity. Literature is missing on this? Provide some context regarding this. The following publications may help:

Anarfi, J. K., & Owusu, A. Y. 2011. “The Making of a Sexual Being in Ghana: The State, Religion and the Influence of Society as Agents of Sexual Socialization.” Sexuality & Culture 15: 1–18.

Fiaveh, D. Y., Izugbara, C. O., Okyerefo, M. P. K., Reysoo, F., & Fayorsey, C. K. 2015. Constructions of masculinity and femininity and sexual risk negotiation practices among women in urban Ghana. Culture, Health & Sexuality, 17(5), 650-662.

Fiaveh, D. Y., Okyerefo, M. P. K., & Fayorsey, C. K. 2015. Women’s experiences of sexual pleasure in Ghana. Sexuality & Culture, 19(4), 697-714.

Response

Thank you for the references, we have added content to the introduction and discussion sections as stated above in comment #1

Comment 3

The subsection on study setting provides little context to the study. For example, how do population size and per capita income relate to FSD? These were not evident in the demographic attributes presented in Table 1 under the results. 

Response 

The details on study setting (population size and per capita income on page 8) are to geographically orient and contextualize the study to all non-Ghanaian and Ghanaian readers. Though this may not be directly related to FSD, we believe this information is important, and relevant to the PlosOne readership.

We are happy to modify with Dr Fiaveh’s recommendations. 

Comment 4

The issues that matter (age, relationship/love, educational attainment, religion, employment status, etc.) were not emphasised in the review/introduction. Yet, they appear instrumental in the analysis/results and discussion. 

Response

We modified the introduction to account for these factors as noted in response to comment 1. Please keep in mind that our project was not meant to be a theoretical, anthropological undertaking. Rather, our project was informed by such published work, and sets the stage for exploring possible solutions to the issue of FSD in patient populations that we, as providers, are faced with in our everyday interactions with our patients. 

Comment 5

For example, the study found a significant relationship between a woman's age and FSD outcome. What does this mean? Would you suggest that younger (say those below age 40) women have different SDs from older women (those above 40 but below 60)?

Response

We believe this issue of Age and FSD was addressed in the discussion as noted below (pages 21-22 of Discussion). As providers, it was important to find out from conducting this work, that lubrication disorders were the most severe form of FSD for “young” women in the study. This finding is important as the clinical “intervention” may be as simple as providing lubricants to our patients for whom this is an issue, to decrease the dysfunction. This is an important finding from a clinical/practitioner/implementation science perspective. 

The following can be found on pages 23-24 of the Discussion:

“It is often assumed that sexual dysfunctions in women are associated with increasing age, the onset of menopause, and several associated hormonal changes [5, 6, 13, 17-22]. However, our results show that FSD is quite common in premenopausal women, particularly those between the ages of 18 and 39 years. Women in this age group were most likely to report that their FSD was severe, with lubrication disorders being the most common disturbance…

The root cause of FSD for these “young” women may have less to do with age and/or biological changes in hormonal regulation, and are most likely attributable to the physical/mechanical stress imposed on the vaginal tissue during sexual encounters. For women, vaginal lubrication is an important part of sex [23]. Lubrication readies the vagina for penetration, and reduces any accompanying friction or irritation during intercourse [4, 23, 24]. Without adequate lubrication, the frictional force imposed on the delicate vaginal tissue can result in bruising and tearing [23], which may lead to infections, painful sexual experiences and overall dissatisfaction with sex. Not surprisingly, lubrication, pain and satisfaction were the three FSD domains most associated with FSD severity in our study. And although sexual pain was the only statistically significant predictor of FSD-associated care seeking behavior, likely at the root of the woman’s dyspareunia is inadequate lubrication. Though lubrication usually occurs naturally, some women become more lubricated than others [23]. It is important that women and their partners understand the role that lubrication plays in comfortable intercourse to reduce the woman’s susceptibility for infections, as well as their painful sexual experiences. More studies are needed to evaluate the access, utilization and perception of personal lubricants amongst Ghanaian women and couples. 

Comment 6 

Some more details about the methodology is required—ethical protocol and procedure/techniques.

Response

We added the following to the methods section (page 10): 

“As specified and approved by the University of Health and Allied Sciences’ Research Ethics Committee, we obtained written consent from all participants to conduct this study. All consent forms were kept separately from the de-identified survey questionnaires, in a secure location under the supervision of the final author. Keeping the signed consents separate from the survey questionnaires was to ensure that participants’ responses could not be connected to their identities.”

Comment 7

Was the survey face-to-face? 

Response 

As noted on page 11: 

“All surveys were interviewer led and completed in-person with each respondent.”

We have added “face-to-face” to be more specific

“All surveys were interviewer led and completed in-person (i.e. face-to-face) with each respondent.”

Comment 8 

Under what condition where the questionnaires administered? 

Response 

We have added the following on page 12: 

“All interviews were conducted in a screen-protected and discrete study-designated area of the outpatient departments of the Ho Teaching Hospital where the study was conducted… We pretested the survey in the following two steps: i) Interviewing of random respondents within the Ho Teaching Hospital to determine participants’ language preferences and to ensure uniformity among study personnel, and 2) under the supervision of the senior author (BI), interviewers practiced administering the survey to ensure consistency of delivery and translation of questions by all study personnel.

Comment 9 

More clarity about the local dialect used since Ewe is not the only local language in the Volta region of Ghana. How were translations/transliterations done? Sexuality is a sensitive issue across ‘borders’ and since the questionnaires were administered by trained study personnel, further details about the processes/techniques could be useful. 

Response

We added the following to the document (page 12) 

“Though there are many languages spoken in the Volta region, from our training and pre-testing phases preparatory to the study, we found most respondents could comfortably speak Ewe, Twi, English or combinations of these. We pretested the survey in the following two steps: i) Interviewing of random respondents within the Ho Teaching Hospital to determine participants’ language preferences and to ensure uniformity among study personnel, and 2) interviewers practice administering the survey to other trainees to ensure consistency of delivery and translation of questions by all study personnel.”

Comment 10 

Another point that deserves much clearer attention and may be an interesting facet of the point above are the measurements— variables. How do relationship status or residential setting or religious affiliation directly impact FSD? And how do we measure these? On the basis of some typology developed, i.e., using a questionnaire?

Response 

We hope the enhancements we added to the introduction and discussion pages of the manuscript (Reponses to Comment #1) address this comment as well. We have also added the approved questionnaire we used in the study as an appendix. We reference the reader to see the appendix as noted below and on page 10 of the manuscript: 

“Data Collection Instrument & Measures 

Demographics

A standard questionnaire (see Appendix) was used to collect socio-demographic data about participants’ demographic …”

 Comment 11

For example, how would different units of analysis in the sample, e.g., pregnant women, non pregnant women, women with kids/children and/or certain number of children, and non-heterosexual women impact the results/claims? How does sexuality operate in Ghanaian society that might silence or closet FSD? Does compulsory heterosexuality (penovaginal 'penetrative' sex) rely upon FSF/D? 

Response: 

In addition to the changes made to the introduction and discussion (comment #1) to address these comments, We have also added the following to the Discussion section on the study’s limitations section (pages 28-29) to address this comment in particular: 

“How does sexuality operate in Ghanaian society that might silence or closet FSD?” 

 “Finally, our study does not account for differences in FSD outcomes based on the women’s pregnancy status or sexual preferences. Though the FSFI was designed to be used non-discriminately, and has in fact been used to evaluate FSD outcomes in pregnant and non-pregnant heterosexual, lesbian and bisexual populations [13, 25-27], because we do not explicitly compare outcomes within these subgroups in our study, the findings should be interpreted cautiously. As is common in most sexuality studies, our results may be more reflective of a heterosexual majority and/or those with less conservative sexual attitudes and values [4, 28], than that of sexual minorities and/or others with conservative sexual attitudes. This is particularly salient in a country like Ghana, where overt discussions of sexual matters are frowned upon [15]. Those who are most reticent about sexual issues may be most vulnerable to adverse sexual outcomes, and may require greater clinical interventions. Thus future studies should attempt to evaluate FSD outcomes in pregnant Ghanaian women, as well as non-heterosexual individuals, who may have these and other unique challenges. Future research endeavors should also consider qualitative research methodologies that will better contextualize and support the study findings, through the use of narratives and theoretical frameworks.”

Comment 12: 

The FSFI questionnaire has major flaws in 'todays' Ghana. Since the paper is on rural agrarian setting in a surfer of Ghana, a section on limitation will be useful.

Response: 

Since its inception in 2000 by Rosen et al, the FSFI has been used extensively to evaluate sexual outcomes in a variety of female populations as noted in our comment above (pregnant, non-pregnant, heterosexual, lesbian, bisexual, etc etc). We selected that instrument because 1) it does not require individuals to be in heterosexual relationships (unlike the Golombok-Rust Inventory of Sexual Satisfaction (GRISS)) which was used in the Amidu study in Ghana [29, 30]; 2)it has been used in several culturally sensitive populations [25, 31-33], and is not explicit (like the Derogatis Interview for Sexual Function-Self Report (DISF-SR) and others)[28]; 3) it allows for comparison of the results across the published literature, with other African and non-African populations [7, 13], and 4) most importantly, the instrument has clinical relevance and conforms with the DSM-V classifications; which we, as practitioners, deem important [34]. 

Preparatory to the research, as noted in the manuscript (comment #9), we pre-tested the instrument with patients to assess their comfort, we ensured that study personnel were also consistent in the survey administrations, we consulted with various other Ghanaian quantitative researchers in this field (Samuel Oppong [35]), CharlesTakyi (Erica Goldstein and Charles Takyi, Female Sexual Dysfunction:Prevalence and Risk in Ghana. November 2015: Physicians For Human Rights National Student Conference) prior to selecting the FSFI as the most appropriate tool for our study’s purpose and clinical/implementation science agenda. 

Interestingly, we found no issues with using this instrument in today’s Ghana, particularly with our Volta participants who we would have expected to be less forthcoming than those in more urban areas like Accra or Kumasi. Our pre-test activities informed us well on how the study would be received-and it was a positive one. Out of 407 women approached, 83.8% agreed to participate in the study. 

Questions in the FSFI seem to have transcended time and populations. Today’s Ghana is less conservative, with the portrayal of Ghana’s female sexuality on shows such as Christiane Amanpour’s sex and love which was viewed all over the world (https://qz.com/africa/1254088/cnn-ghana-and-moesha-boduong-spark-backlash-on-sex-love-and-african-women/), women like Akumaa Mama Zimbi; Ghana’s “Sex Doctor”, openly discussing “bedroom wahala” on radio, television and social media. 

We kindly request Dr Fiaveh to please clarify this comment as we may be misunderstanding what he means by the instrument has “major flaws in 'todays' Ghana”. For ease of reference, we have included the survey instrument in the Appendix of the manuscript on pages 28-32. 

Comment 13 

The items measured, i.e., desire, arousal, lubrication, orgasm, satisfaction, and pain during sexual intercourse do not suffice to make generalizations. If women desire pleasurable and safe sex, should’t we be discussing/asking what role sexual preference/orientation (e.g., oral sex, use of condom) play in this? What about sex positions, romance, the penis; size, length, girth, etc., for those who prefer 'penetrative' sex play in FSD? If not why not?

Response

We believe that Dr Fiaveh has covered much of these in his work [36, 37] 

(Daniel Yaw Fiaveh, Condom Myths and Misconceptions: The Male Perspective. Global Journal of Medical research, 2012. 12(5).

Fiaveh, D.Y. and Michael PK Okyerefo, Femininity, Sexual Positions and Choice. Sexualities, 2019. 22(1-2): p. 131-147.

Fiaveh, D.Y. Phallocentricism, female penile choices, and the use of sex toys in Ghana https://doi.org/10.1177/1363460718781975)

and though we used some of this information to inform our work, our focus was not only to identify the pathological prevalence, but to determine the types of FSD (desire, arousal, lubrication, orgasm, satisfaction, and pain), and their severity from a clinical standpoint so we can incorporate these findings in clinical practice.

We acknowledge that these outcomes may not be generalizable nor are they comprehensive. However, we designed our study with the FSFI instrument to assist us in practically determining the prevalence of the various types of FSD designated in the DSM-V, in order to treat the distressed patient who seeks our care.

Penis size, sexual positions, oral sex and other aspects of sexual behavior/preferences though important, are not as important for us, as epidemiologists, clinicians and implementation scientists, as finding that (Page 5): 

“Over a quarter of the sample (27.6%, n=40) met the cut-off for moderate to severe FSD. In age-adjusted models, lubrication disorder was associated with 45 times the odds of moderate/severe FSD (95% CI: 9.27, 264.5; p<0.001), pain with 17times the odds (OR: 17.18, 95% CI: 4.50, 65.50; p<0.001) and satisfaction almost 5times the odds (OR: 4.69, 95% CI: 1.09, 20.2; p=0.04)…“ 

Such a salient finding allows us to recognize the potentially debilitating consequences of the dysfunction on our patients. As an example, knowing that lubrication disorders are the most oppressive allows us to explore treatment options for the patient-options which, as previously mentioned, may be “simple” to implement. Relieving the frictional force imposed on the delicate vaginal tissue during penetrative intercourse can alleviate bruising and tearing, and reduce the woman’s susceptibility to infections, painful sexual experiences and overall dissatisfaction with sex.

Oral sex, condom use, sex positions, romance, the penis; size, length, girth, etc., for those who prefer 'penetrative' sex play etc. though important, would not sufficiently treat a woman’s lubrication disorder. Treating that disorder, however, will enhance the woman’s sexual experience each time she encounters her “preferred” penis: size, length, girth, etc; preferred sexual position; or preferred romancing. 

Additionally, designing a study that focused primarily on penis preference and condom use would have been presumptive on our part, restricting the study’s generalizability to participants with heterosexual preferences solely; an issue that Dr Fiaveh has already noted would not have been favorable; and that we have attempted very hard to avoid. 

Comment 14

The results reflect the methods employed in the data gathered. Narratives were missing and this is ‘sad’. There is need for some qualitative data, interviews, to support positivists’ stance and claims being made in the paper including the proposal to develop contextually appropriate interventions for the subpopulation studied in Ghana. 

Response: 

We have added the following to the discussion (page 29)

“Future research endeavors should also consider qualitative research methodologies that will better contextualize and support the study findings, through the use of narratives and theoretical frameworks. “

Comment 15 

Would the results differ among Akan groups/speakers?

Response

We sought to conduct this study in women in the Volta region because none of the existing literature has evaluated these outcomes in this region of Ghana. The six region study by Ibmeah and co [35] claimed to evaluate outcomes in a “representative” part of Ghana, but omitted Volta participants. The study included female participants from Greater Accra, Ashanti, Western, Central, Brong-Ahafo and Eastern regions but excluded the Volta region. Part of our agenda was to fill a gap in the literature and determine if findings from these 6 regions differ from that of Volta participants. What we found was that our findings are similar. Our estimate of 48.3% approximated the larger, more inclusive study’s estimate of 45.6%

We have modified the sentence on page 22 of the manuscript to be more specific: 

“ The FSD prevalence estimate of 48.3% in our study, approximates the global prevalence of 40.9% [7] and is similar to rates reported for non-clinical, mostly urban Ghanaian females in heterosexual relationships (45.6%) who hailed from the Greater Accra, Ashanti, Western, Central, Brong-Ahafo and Eastern regions. [35].” 

Comment 16 

While the discussion is logically organised, it is underdeveloped probably due to the absence of a theory/framework to hold the arguments together. The contribution of this paper will be better appreciated if it were supported with a theory/framework/model.

Response: 

Our study’s aim, cited on page 6 of the manuscript, were to describe the prevalence and severity of FSD amongst a group of Ghanaian women in the outpatient setting of the predominantly rural Volta region of Ghana; and to describe the predictors of FSD severity and the care seeking behaviors of women with the condition. 

In the results section (pages 13-21), we fulfill these aims by reporting the prevalence of FSD, the severity of the dysfunction and the care-seeking behaviors of our study participants. 

 We have also added this comment as an addition to the limitations section: 

“Future research endeavors should also consider qualitative research methodologies that will better contextualize and support the study findings, through the use of narratives and theoretical frameworks. The contribution of this paper may be better appreciated if it were supported with qualitative theories/frameworks/models (page 26).”

Comment 17 

The part about the link to rural residential setting and implications for FSD (see paragraph 1 line 9-12 under Discussion) is contested and can constitute a sensitive research design including stereotype. Is this a finding or an existing literature? If this is so, this should also be interrogated. This could be due to negative skewness in the sampling technique. You need to engage the data a bit more closely and situate your argument within a paradigm.

Response: 

The statements in lines 9-12 of the discussion have been substantiated across the literature (including calls for intervention by the Ghana Ministry of Health for studies to improve outcomes in rural populations (Ministry of Health Ghana. Ghana Millenium Development Goals Acceleration Framework, 2015 Strategy and Operational Plan. Accra, Ghana2015.), other Ghanaian researchers

Amalba, A., et al. (2018). "Working among the rural communities in Ghana - why doctors choose to engage in rural practice." BMC Med Educ 18(1): 133. 

Development Initiatives 2018 (4/5/205). "Ghana’s rural savannah areas suffer persistently high poverty." Retrieved 8/1/2019, from http://devinit.org/post/ghanas-rural-savannah-areas-suffer-persistently-high-poverty/.

Bazzano, A.N., B. Kirkwood, C. Tawiah-Agyemang, S. Owusu-Agyei, and P. Adongo, Social costs of skilled attendance at birth in rural Ghana. Int J Gynaecol Obstet, 2008. 102(1): p. 91-4.

Kyei-Nimakoh M, Carolan-Olah M, McCann TV. Millennium development Goal 5: progress and challenges in reducing maternal deaths in Ghana. BMC pregnancy and childbirth. Mar 9 2016;16:51.)

 And the American College of Obstetrics and Gynecology, the governing body for

Obstetricians and Gynecologists, the head of which is currently Dr Adanu, who, along

with Prof J.K Anarfi, J. K published one of the earliers papers on the Ghana Ministry of

Health’s restrictive Sexual and Reproductive Health Strategy[16] )

It is also assuring that our study findings support the statements regarding participants in our study who described themselves as residing in rural setting. As we note in the discussion (page 14):

“Approximating the rural population of the Volta Region, over a third of the FSD women resided in rural settings (37.9% vs 20.6% p=0.001) and tended to be multiparous, with a significantly greater proportion having at least three children (31.7% vs 18.1; p=0.033), compared to those without FSD.”

On a grander and more relevant scale, women in the Volta region have been traditionally missing from the Ghana FSD literature (see comment 15 above) and this study offers a unique opportunity for us, as implementation and research scientists to work with an under-represented population. Our sampling strategy was designed to answer the research question, and not intentionally skewed. We see our study design as an opportunity to give voice to those who have been omitted from the literature, rather than a stereotyping. We sought to ask a very specific research question (Page 8) 

“In this paper, we describe the prevalence and severity of FSD amongst a group of Ghanaian women in the outpatient setting of the predominantly rural Volta region of Ghana. We describe the predictors of FSD severity and the care seeking behaviors of women with the condition; in the hopes of identifying possible opportunities for developing contextually appropriate interventions for this subpopulation of women.

And on pages 14-21 we answer those questions. 

We implore Dr Fiaveh to see the gap that this study is filling, and the opportunity it presents for offering solutions for women who tend to be dismissed from such work. The large, six-region study by Imbeah and co [35] even excluded Volta residents. 

As we state in the Discussion (page 21): 

“This is the first study that to our knowledge, evaluates the prevalence, severity and help-seeking behaviors of women with FSD in the mostly rural and agrarian setting of the Volta region of Ghana.”

Comment 18

There are language problems and sloppiness. The authors need to proofread the paper thoroughly.

Response

This comment is in direct contraction to the reviewer’s response to item 4 on the review (see below). We ask Dr Fiaveh to please highlight such issues for our attention and correction. 

4. Is the manuscript presented in an intelligible fashion and written in standard English?

Reviewer #1: Yes

Reviewer #2: Yes

Comment 19

All my very best!

Response:

The honor is ours. We are grateful for your time

 

Reviewer #2: 

I am impressed by this manuscript, both by the topic it studies, which I am sure is under studied in all settings, but especially in low resource settings such as Ghana, and by the quality of the paper.

Response

Thank you for recognizing the amount of work we put into this work, which is near and dear to our hearts. 

I have just a few comments which are necessary to address before this paper can be published.

Comment 1 

1. The methods section doesn't ever state which inferential statistics are performed. The regression analyses need to be explained.

Response

We discuss the univariate and bivariate analyses but realize that we did not touch on the regression models in the statistical analyses section. Thank you for noticing that. 

We have added the following to the statistical analyses section on page 13:

“Age was found to be associated with several other covariates and predictors, and was the most significant predictor of FSD outcome in this study. Adjusting for age only in logistic regression analyses gave us the most parsimonious models. Thus in our logistic regression models, we present both crude and age-adjusted effect estimates. All computations were done at the 95% confidence level, and we report odds ratios and their associated 95% confidence intervals. We used p <0.05 to determine statistical significance for all analyses.”

Comment 2: 

2. Did interviewers only ask the questions about care-seeking to those women who answered in the affirmative about having symptoms of FSD?

Response

Care seeking behaviors were asked of all participants. Those who sought care (responded “Yes” to question c1) were asked where (c3) and why (c4) they sought help. Those who did no seek help (“No” to question c1 were asked why they did not seek help (c2). We have included the survey instrument in the appendix to orient the reader. We have also updated Table 3 (pages 20-21) with footnotes to orient the reader: 

Part c

C1.Did you seek for help when you experienced the female sexual dysfunction?

Yes [ ] No [ ]

C2. If no to C1, why didn’t you seek for help?

Health provider cannot help me [ ] time constraints [ ] 

Others…………………………………………………..

c3. If yes to C1, where did you seek help from? 

Gynecologist ( ) general practitioner ( ) Herbalist ( ) friends ( ) family ( ) Prayer camp ( )

Others………………………

C4. If yes to C1, why did you seek help?

Advised to by relatives/friends/pastor [ ] Referred from a formal healthcare personnel [ ] condition is causing relationship/marriage strains [ ] condition is causing emotional strains [ ] 

Others……………………………………………………………………………………………………… 

Have also added the following legend to table 3

*asked of those who answered “Yes” to the question “Did you seek for help when you experienced the female sexual dysfunction?” (see Appendix)

**asked of those who answered “No” to the question “Did you seek for help when you experienced the female sexual dysfunction?” (see Appendix)

Comment 3

3. It is written, "amongst women with FSD, the most commonly endorsed domains were..." What is meant by "endorsed"?

Response

Endorsed in this context is used for those who met the cutoff for the particular disorder. Thus women who endorsed pain disorders on Table 1 for example, met the cut-off for falling into the pain FSD category. Happy to replace if the reviewer could make a recommendation. 

Comment 4

4. Why is Table 3 only unadjusted estimates? Why aren't there multivariate analyses conducted on these data?

Response

We have updated this table with adjusted estimates, similar to Table 2. We also took this an opportunity to make relevant changes to the Results section, both text and tables. 

Comment 5

5. I would caution the authors against the causal language at the beginning of the discussion section. "This is the first study...that evaluates the predictors...." These are associations, not predictors, since the data are all cross-sectional.

Response

The sentence has been updated as follows: 

This is the first study that to our knowledge, evaluates the prevalence, severity and help-seeking behaviors of women with FSD in the mostly rural and agrarian setting of the Volta region of Ghana.

Comment 6

6. The end of the first paragraph in the discussion section clearly lay out the problems that SD poses for women. Is it possible to state a few interventions that could address these? The authors go into some of them later in the discussion section, but it feels like that could be a good place for some information as well.

Response

We added the following on page 23: 

“The Ministry of Health’s reproductive health policy could be expanded beyond the prevention of unwanted pregnancies and sexually transmitted infections [15, 16, 30] to more holistically encourage providers to discuss sexual health and sexual dysfunctions (both male and female forms) with their patients. We acknowledge that such a recommendation would require an enormous psycho-socio-cultural shift across all facets of the country’s healthcare system, but without such a strategy, it will be difficult to recognize the discriminant impact that sexual dysfunctions have on some of the most vulnerable members of Ghanaian society.”

Comment 7

7. While I do not disagree that a more gender equitable physician workforce would likely help with this, and other, issues (and I appreciate very much the authors going into some detail as to the advantages of having a gender equitable physician workforce), a) that is going to take a very long time to achieve, and b) all physicians *should* be able to address the health concerns of their patients. The authors do say that there is a need to improve "social empathy, assessment and management", but don't say if there are other ways that have proven effective to help physicians help their patients.

Response

We have added the following to page 27: 

Because of the social stigma around female sexuality, women tend to avoid and/or are embarrassed to discuss their sexual health with their health care professionals (HCPs) [38]. HCPs-both physicians and APPs-can be trained to improve their empathetic responses to their patients’ sexual frustrations by initiating (and maintaining) a sexual health conversation in a manner that is comfortable for the woman to convey her concerns. This initial step is crucial to the trust building necessary for the trained provider to assess the patient, diagnose, treat and manage their FSD [38-40]. These tools-educating women, training HCPs, and providing communication tools to HCPs-may serve to decrease the stigma associated with sexual health discussions, and facilitate the treatment-oriented dialogue between patients and their providers when dealing with FSDs [38].

Comment 8 

8. In addition to what I wrote above, the authors also did not mention nurses or midwives, who are both far more plentiful in the country, and far more likely to be female. Is there a role for nurses and / or midwives in this space?

Response 

This is a great point, Thank you. The following has been added on page 26: 

“Transitioning to this model of care however, may take a long time to achieve, as it requires not only an attitudinal and cultural shift in the existing healthcare infrastructure, but may also require a dismantling of pre-existing institutional and structural barriers. Bridging this transition, may be a healthcare model that utilizes Advanced Practice Providers (APPs) such as nurse midwives, who often serve as primary caregivers for patients with obstetric and gynecological needs. Not only is the provider:patient ratio higher for these APPs, but the gender balance between patients and providers is not as disproportional. Nurse midwives could be trained to screen for FSD as part of their “routine” patient care; relieving the workload expectations of other providers.“

1. World Health Organization. Sexual and reproductive health: Defining Sexual Health. 2006a 7/16/2019]; Available from: https://www.who.int/reproductivehealth/topics/sexual_health/sh_definitions/en/.

2. Sandfort, T.G. and A.A. Ehrhardt, Sexual health: a useful public health paradigm or a moral imperative? Arch Sex Behav, 2004. 33(3): p. 181-7.

3. Association, A.P., Highlights of Changes from DSM-IV-TR to DSM-5. 2013.

4. Richard D. McAnulty and M. Michele Burnette, Sex and Sexuality: Sexual function and dysfunction. 2006, Greenwood Press: Westport, CT [u.a.].

5. Nappi, R.E. and M. Lachowsky, Menopause and sexuality: prevalence of symptoms and impact on quality of life. Maturitas, 2009. 63(2): p. 138-41.

6. Nappi, R.E., L. Cucinella, S. Martella, M. Rossi, L. Tiranini, and E. Martini, Female sexual dysfunction (FSD): Prevalence and impact on quality of life (QoL). Maturitas, 2016. 94: p. 87-91.

7. McCool, M.E., A. Zuelke, M.A. Theurich, H. Knuettel, C. Ricci, and C. Apfelbacher, Prevalence of Female Sexual Dysfunction Among Premenopausal Women: A Systematic Review and Meta-Analysis of Observational Studies. Sex Med Rev, 2016. 4(3): p. 197-212.

8. Smith, N.K., J. Madeira, and H.R. Millard, Sexual function and fertility quality of life in women using in vitro fertilization. J Sex Med, 2015. 12(4): p. 985-93.

9. Smith, L.J., J.P. Mulhall, S. Deveci, N. Monaghan, and M.C. Reid, Sex after seventy: a pilot study of sexual function in older persons. J Sex Med, 2007. 4(5): p. 1247-53.

10. Flynn, K.E., L. Lin, D.W. Bruner, J.M. Cyranowski, E.A. Hahn, D.D. Jeffery, J.B. Reese, B.B. Reeve, R.A. Shelby, and K.P. Weinfurt, Sexual Satisfaction and the Importance of Sexual Health to Quality of Life Throughout the Life Course of U.S. Adults. J Sex Med, 2016.

11. Safarinejad, M.R., Female sexual dysfunction in a population-based study in Iran: prevalence and associated risk factors. Int J Impot Res, 2006. 18(4): p. 382-95.

12. Laumann, E.O., A. Paik, and R.C. Rosen, Sexual dysfunction in the United States: prevalence and predictors. JAMA, 1999. 281(6): p. 537-44.

13. West, S.L., L.C. Vinikoor, and D. Zolnoun, A systematic review of the literature on female sexual dysfunction prevalence and predictors. Annu Rev Sex Res, 2004. 15: p. 40-172.

14. Rahman, S., Female Sexual Dysfunction Among Muslim Women: Increasing Awareness to Improve Overall Evaluation and Treatment. Sex Med Rev, 2018. 6(4): p. 535-547.

15. John Kwasi Anarfi and Adobea Yaa Owusu, The Making of a Sexual Being in Ghana: The State, Religion and the Influence of Society as Agents of Sexual Socialization. Sexuality & Culture, 2010. 15(1): p. 1-18.

16. Adanu, R.M., J. Seffah, J.K. Anarfi, N. Lince, and K. Blanchard, Sexual and reproductive health in Accra, Ghana. Ghana medical journal, 2012. 46(2): p. 58-65.

17. Coelho, G., C. Frange, M. Siegler, M.L. Andersen, S. Tufik, and H. Hachul, Menopause Transition Symptom Clusters: Sleep Disturbances and Sexual Dysfunction. J Womens Health (Larchmt), 2015. 24(11): p. 958-9.

18. Prairie, B.A., S.R. Wisniewski, J. Luther, R. Hess, R.C. Thurston, K.L. Wisner, and J.T. Bromberger, Symptoms of depressed mood, disturbed sleep, and sexual problems in midlife women: cross-sectional data from the Study of Women's Health Across the Nation. J Womens Health (Larchmt), 2015. 24(2): p. 119-26.

19. Makara-Studzinska, M.T., K.M. Krys-Noszczyk, and G. Jakiel, Epidemiology of the symptoms of menopause - an intercontinental review. Prz Menopauzalny, 2014. 13(3): p. 203-11.

20. Bachmann, G.A. and S.R. Leiblum, The impact of hormones on menopausal sexuality: a literature review. Menopause, 2004. 11(1): p. 120-30.

21. Bachmann, G.A., Influence of menopause on sexuality. Int J Fertil Menopausal Stud, 1995. 40 Suppl 1: p. 16-22.

22. Remez , L. Multiple Factors, Including Genetic and Environmental Components, Influence When Menopause Begins. 8/9/2016]; Available from: https://www.guttmacher.org/about/journals/psrh/2001/09/multiple-factors-including-genetic-and-environmental-components.

23. International Society for Sexual Medicine. Why is vaginal lubrication important for women? 7/31/2019]; Available from: https://www.issm.info/sexual-health-qa/why-is-vaginal-lubrication-important-for-women/.

24. Jennifer Huizen for Medical News Today. Causes and treatment of vaginal cuts. May 2nd, 2019 7/31/2019]; Available from: https://www.medicalnewstoday.com/articles/325100.php.

25. Khalesi, Z.B., M. Bokaie, and S.M. Attari, Effect of pregnancy on sexual function of couples. Afr Health Sci, 2018. 18(2): p. 227-234.

26. Duarte-Guterman, P., B. Leuner, and L.A.M. Galea, The long and short term effects of motherhood on the brain. Front Neuroendocrinol, 2019. 53: p. 100740.

27. Breyer, B.N., J.F. Smith, M.L. Eisenberg, K.A. Ando, T.S. Rowen, and A.W. Shindel, The impact of sexual orientation on sexuality and sexual practices in North American medical students. J Sex Med, 2010. 7(7): p. 2391-400.

28. Anto-Ocrah, M., J. Bazarian, V. Lewis, C.M. Jones, T.A. Jusko, and E. Van Wijngaarden, Risk of female sexual dysfunction following concussion in women of reproductive age. Brain Inj, 2019. 33(11): p. 1449-1459.

29. Amidu, N., W.K. Owiredu, C.K. Gyasi-Sarpong, E. Woode, and L. Quaye, Sexual dysfunction among married couples living in Kumasi metropolis, Ghana. BMC Urol, 2011. 11: p. 3.

30. Amidu, N., L. Quaye, A.A. Afoko, P. Karikari, B.B. Gandau, E.O. Amoah, and E. Nuwoku, Golombok Rust Inventory of Sexual Satisfaction for the presence of sexual dysfunction within a Ghanaian urological population. Int J Impot Res, 2014. 26(4): p. 135-40.

31. Agustus, P., M. Munivenkatappa, and P. Prasad, Sexual Functioning, Beliefs About Sexual Functioning and Quality of Life of Women with Infertility Problems. Journal of human reproductive sciences, 2017. 10(3): p. 213-220.

32. Fakhri, A., A.H. Pakpour, A. Burri, H. Morshedi, and I.M. Zeidi, The Female Sexual Function Index: translation and validation of an Iranian version. J Sex Med, 2012. 9(2): p. 514-23.

33. Rehman, K.U., M. Asif Mahmood, S.S. Sheikh, T. Sultan, and M.A. Khan, The Female Sexual Function Index (FSFI): Translation, Validation, and Cross-Cultural Adaptation of an Urdu Version "FSFI-U". Sexual medicine, 2015. 3(4): p. 244-250.

34. Isidori, A.M., C. Pozza, K. Esposito, D. Giugliano, S. Morano, L. Vignozzi, G. Corona, A. Lenzi, and E.A. Jannini, Development and validation of a 6-item version of the female sexual function index (FSFI) as a diagnostic tool for female sexual dysfunction. J Sex Med, 2010. 7(3): p. 1139-46.

35. Emelia P Imbeah, Barima A Afrane, Irene A Kretchy, Joseph A Sarkodie, Franklin Acheampong, Samuel Oppong and Patrick Amoateng,,, Prevalence and Self-Management of Female Sexual Dysfunction among Women in Six Regions of Ghana: A Cross-Sectional Study. J Womens Health Issues Care, 2015. 4(6).

36. Daniel Yaw Fiaveh, Condom Myths and Misconceptions: The Male Perspective. Global Journal of Medical research, 2012. 12(5).

37. Fiaveh, D.Y. and Michael PK Okyerefo, Femininity, Sexual Positions and Choice. Sexualities, 2019. 22(1-2): p. 131-147.

38. Kingsberg, S.A., J. Schaffir, B.M. Faught, J.V. Pinkerton, S.J. Parish, C.B. Iglesia, J. Gudeman, J. Krop, and J.A. Simon, Female Sexual Health: Barriers to Optimal Outcomes and a Roadmap for Improved Patient-Clinician Communications. J Womens Health (Larchmt), 2019. 28(4): p. 432-443.

39. Feldhaus-Dahir, M., Female sexual dysfunction: barriers to treatment. Urol Nurs, 2009. 29(2): p. 81-5; quiz 86.

40. Gott, M., E. Galena, S. Hinchliff, and H. Elford, "Opening a can of worms": GP and practice nurse barriers to talking about sexual health in primary care. Fam Pract, 2004. 21(5): p. 528-36.

---

## [Decision Letter · Decision Letter 1]

7 Nov 2019

PONE-D-19-22745R1

“I did not know it was a medical condition”: Predictors, Severity and Help Seeking Behaviors of Women with Female Sexual Dysfunction in the Volta Region of Ghana”

PLOS ONE

Dear Dr Anto-Ocrah,

Thank you for submitting your manuscript to PLOS ONE. After careful consideration, we feel that it has merit but does not fully meet PLOS ONE’s publication criteria as it currently stands. Therefore, we invite you to submit a revised version of the manuscript that addresses the points raised during the review process.

We would appreciate receiving your revised manuscript by Dec 22 2019 11:59PM. To enhance the reproducibility of your results, we recommend that if applicable you deposit your laboratory protocols in protocols.io, where a protocol can be assigned its own identifier (DOI) such that it can be cited independently in the future. For instructions see: http://journals.plos.org/plosone/s/submission-guidelines#loc-laboratory-protocols

We look forward to receiving your revised manuscript.

Kind regards,

Joshua Amo-Adjei, Ph.D

Academic Editor

PLOS ONE

Reviewers' comments:

Reviewer's Responses to Questions

**Comments to the Author**

1. If the authors have adequately addressed your comments raised in a previous round of review and you feel that this manuscript is now acceptable for publication, you may indicate that here to bypass the “Comments to the Author” section, enter your conflict of interest statement in the “Confidential to Editor” section, and submit your "Accept" recommendation.

Reviewer #1: All comments have been addressed

2. Is the manuscript technically sound, and do the data support the conclusions?

Reviewer #1: Yes

3. Has the statistical analysis been performed appropriately and rigorously? 

Reviewer #1: Yes

4. Have the authors made all data underlying the findings in their manuscript fully available?

Reviewer #1: Yes

5. Is the manuscript presented in an intelligible fashion and written in standard English?

Reviewer #1: Yes

6. Review Comments to the Author

Reviewer #1: Dear Authors,

I have read your revised paper and it reads better than the previous one. I feel the contribution of this study will be better appreciated if it were grounded in a theory/model.

Minor issues:

• The sentence on sexual problems in response to Comment 1 in the introduction is long. You may want to reconsider for clarity.

• Since the term human sexuality is introduced in the work for the first time, I am not quite sure about the appropriateness of the current location, i.e., discussion (pages 21-22). In my view, it will better fit for the introduction.

• A recap of the main objective(s) in the first paragraph of the conclusion would be useful.

• It’s surprising the authors find the suggestion to proofread the paper for clarity a contradiction to other segments of the online reviewer form. Overall assessment of an article (in relation to the intelligibility and writing standard) is not coterminous with specific errors noted in same.

I want to commend the authors for the point-by-point responses to the specific issues raised in the article and I look forward to reading the published version. Thank you for your fine contribution.

All my very best!

7. PLOS authors have the option to publish the peer review history of their article (what does this mean?). If published, this will include your full peer review and any attached files.

Reviewer #1: Yes: Daniel Yaw Fiaveh

---

## [Author Response · Author response to Decision Letter 1]

21 Nov 2019

Comment 

Reviewer #1: Dear Authors,

I have read your revised paper and it reads better than the previous one. I feel the contribution of this study will be better appreciated if it were grounded in a theory/model.

Response

We addressed this comment in the limitations for future studies in the discussion section: 

“Future research endeavors should also consider qualitative research methodologies that will better contextualize and support the study findings, through the use of narratives and theoretical frameworks. The contribution of this paper may be better appreciated if it were supported with qualitative theories/frameworks/models.”

Minor issues:

Comment 

• The sentence on sexual problems in response to Comment 1 in the introduction is long. You may want to reconsider for clarity.

Response

We have made some edits as follows (these are also tracked in the document):

Sexual problems are often multifactorial and can be classified into four broad categories, which are: biomedical (biological factors such as pregnancy, injury and disability, cardiovascular and other chronic diseases that can interfere with intercourse); intrapsychic (psychological elements within the individual, that influence their sexual expression e.g., values learned about one’s body, nudity, “where babies come from”, puberty, etc); interpersonal (factors that influence one’s ability to engage in sexual relationships, such as communication difficulties with a partner, the extent to which partners share compatible visions of sex, eroticism, pleasure etc.); and socio-cultural/economic/political (‘blueprints’ for sexual norms, beliefs, values, practices and attitudes that are created and imposed on an individual based on the ‘moral code’ of their society. Includes religion.) [4, 15]. These four factors often overlap, and rarely occur in isolation to cause the sexual dysfunction [4].

Comment

• Since the term human sexuality is introduced in the work for the first time, I am not quite sure about the appropriateness of the current location, i.e., discussion (pages 21-22). In my view, it will better fit for the introduction.

Response

We revised the discussion to be more specific to sexual function and reads as follows: 

Though human sexual function has requisite biological underpinnings, it is usually experienced as a complex interaction of biological, intrapsychic, interpersonal and socio-cultural/economic/political factors [4, 15].

Comment 

• A recap of the main objective(s) in the first paragraph of the conclusion would be useful.

Response

We updated the conclusion to read as follows: 

The study’s main objective was to describe the prevalence and severity of female sexual dysfunction (FSD) amongst a group of Ghanaian women in the outpatient setting of the predominantly rural Volta region of Ghana. Additionally we determine the predictors of FSD severity and care seeking behaviors of women with the condition.We found that FSD is highly prevalent amongst women in rural settings of the Ghanaian community, with lubrication, pain and satisfaction disorders being the most associated with severity.

Comment 

• It’s surprising the authors find the suggestion to proofread the paper for clarity a contradiction to other segments of the online reviewer form. Overall assessment of an article (in relation to the intelligibility and writing standard) is not coterminous with specific errors noted in same.

Response

We must have misinterpreted the following comment by the reviewer: 

“There are language problems and sloppiness...”

Thank you for the clarification. 

Comment 

I want to commend the authors for the point-by-point responses to the specific issues raised in the article and I look forward to reading the published version. Thank you for your fine contribution.

All my very best!

Response

We appreciate your insights

---

## [Editor Report · Decision Letter 2]

27 Nov 2019

“I did not know it was a medical condition”: Predictors, Severity and Help Seeking Behaviors of Women with Female Sexual Dysfunction in the Volta Region of Ghana”

PONE-D-19-22745R2

Dear Dr. Anto-Ocrah,

We are pleased to inform you that your manuscript has been judged scientifically suitable for publication and will be formally accepted for publication once it complies with all outstanding technical requirements.

With kind regards,

Joshua Amo-Adjei, Ph.D

Academic Editor

PLOS ONE
---

## [Editor Report · Acceptance letter]

23 Dec 2019

PONE-D-19-22745R2 

“I did not know it was a medical condition”: Predictors, Severity and Help Seeking Behaviors of Women with Female Sexual Dysfunction in the Volta Region of Ghana” 

Dear Dr. Anto-Ocrah:

I am pleased to inform you that your manuscript has been deemed suitable for publication in PLOS ONE. Congratulations! Your manuscript is now with our production department. 

With kind regards,

on behalf of

Dr. Joshua Amo-Adjei 

Academic Editor

PLOS ONE